# ProjQ: Project-and-Quantize for Adapter-Aware LLM Compression

Wenya Yu [1]  Chao Zhang [2]  Li Wang [1]  Samson Lasaulce [3]  Merouane Debbah [4]

## Abstract

Post-Training Quantization (PTQ) and Low-Rank Adaptation (LoRA) constitute the standard pipeline for efficient Large Language Model (LLM) deployment. However, applying them sequentially poses a problem: PTQ often leaves behind random noise that is spread out (across the model's weights) in a way LoRA can't easily fix, meaning that LoRA ends up wasting its limited capacity trying to fix uncorrectable noise instead of improving task performance. In this paper, we propose **ProjQ**, a novel framework for constraining quantization noise to the low-rank manifold via orthogonal subspace projection. We derive an efficient alternating algorithm that shapes the quantization noise into a low-rank structure, effectively offloading dominant error components to the subsequent adapter while minimizing the residual error in the orthogonal "uncorrectable" subspace. Our theoretical analysis demonstrates that ProjQ preserves strictly greater model plasticity for downstream tasks compared to standard PTQ. Extensive experiments on LLaMA-2, Qwen2.5 and Qwen3 confirm that ProjQ consistently outperforms existing methods in both quantization error compensation and downstream task fine-tuning, achieving up to $2\times$ lower evaluation loss for compensation and matching the performance of standard 4-bit baselines on language modeling tasks with only 3 bits. The code is available on `https://github.com/yy9301/ProjQ`.

---
[1] School of Mathematics and Statistics, Central South University, Changsha, China [2]Department of Computer Science, Central South University, Changsha, China [3]CRAN, Université de Lorraine, CNRS, Nancy, France [4]6G Research Center, Khalifa University, Abu Dhabi, UAE. Correspondence to: Chao Zhang <chao.zhang@cse.edu>.

*Proceedings of the 43rd International Conference on Machine Learning*, Seoul, South Korea. PMLR 306, 2026. Copyright 2026 by the author(s).

## 1. Introduction

Large Language Models (LLMs) have demonstrated remarkable capabilities across a wide range of natural language processing tasks. To deploy these general-purpose models for specific applications, fine-tuning is essential to adapt their representations to downstream domains. However, full fine-tuning of billion-scale parameters is often prohibitively expensive in terms of storage and computation. Consequently, Parameter-Efficient Fine-Tuning (PEFT) methods, particularly Low-Rank Adaptation (LoRA) (Hu et al., 2022), have become the standard paradigm. By injecting trainable low-rank matrices into the frozen pre-trained weights, LoRA enables efficient adaptation while maintaining model quality, making it a cornerstone of modern LLM deployment.

Despite the efficiency of LoRA, the sheer size of base models remains a bottleneck for deployment on resource-constrained devices (such as on-device LLMs (Zou et al., 2026)). Post-Training Quantization (PTQ) addresses this by compressing weights into low-bit formats (e.g., 4-bit or 2-bit), significantly reducing memory footprint and latency (Frantar et al., 2022). However, aggressive quantization inevitably introduces compression noise, which can severely degrade model performance. This creates a dual burden for the downstream adapter: it must not only capture the semantic shift required for the specific task but also compensate for the quantization error induced by the compression. Standard approaches that perform independent quantization followed by fine-tuning (e.g., QLoRA (Dettmers et al., 2023)) are computationally simple but fail to exploit the synergy between these stages. By treating them independently, such methods distribute quantization error isotropically, often corrupting the specific low-rank subspace that the adapter is best suited to correct. Conversely, prior works such as LoftQ (Li et al., 2023) have attempted to bridge this gap by jointly initializing quantization and adapter parameters. While these methods mitigate initialization discrepancies, they typically operate in the weight space without explicitly accounting for input activations or lack rigorous theoretical guarantees.

In this paper, we address these challenges by rethinking the interaction between quantization and adaptation, shifting the focus from weight-only reconstruction to activation-aware layer output minimization, a strategy proven essential for

preventing degradation in aggressive low-bit PTQ regimes. We argue that while the downstream task shift is unknown at the quantization stage, the structure of the quantization noise is controllable. If we can shape the quantization error to reside primarily within a specific low-rank subspace, the subsequent LoRA adapter can efficiently "absorb" this error, preserving its remaining capacity for learning the task. To this end, we introduce **ProjQ** (Project-and-Quantize), a framework that explicitly aligns the quantization process with the corrective mechanism of low-rank fine-tuning. By formulating the problem as a subspace projection objective on the layer outputs, we ensure that the "uncorrectable" component of the noise is minimized.

Our main contributions are **summarized** as follows:

- We propose the **ProjQ** framework, a novel formulation that produces a fine-tuning friendly quantized base model by explicitly aligning post-training quantization with the corrective capacity of Low-Rank Adaptation via orthogonal projection.

- We propose the **Project-and-Quantize algorithm**, an alternating minimization solver that restricts the quantization noise to a low-rank subspace, ensuring it can be compensated by the subsequent low-rank adapter more easily.

- We provide **theoretical analysis** to prove the efficiency of our approach and its convergence, demonstrating that our projection-based approach strictly preserves more model capacity for downstream tasks compared to standard PTQ.

- We **validate our approach** across diverse benchmarks, demonstrating that ProjQ consistently outperforms baselines in both quantization error compensation and downstream task adaptation, achieving lower evaluation loss and enabling models to match the performance of standard baselines with fewer bits.

## 2. Related Works

### 2.1. Post-Training Quantization

PTQ has become a standard strategy for compressing LLMs due to its efficiency and lack of retraining requirements. The primary goal of PTQ is to reduce the precision of model weights (and potentially activations) while minimizing the degradation of output quality. Early approaches relied on simple rounding techniques, such as Round-to-Nearest (RTN), which often incur significant error in low-bit regimes. To mitigate this, more advanced methods leverage calibration data to optimize the quantization process.

A prominent line of work incorporates second-order information to guide quantization. For instance, GPTQ (Frantar

et al., 2022) utilizes a proxy of the Hessian of the loss function to iteratively update weights, achieving high accuracy in 4-bit and 3-bit settings. Similarly, AWQ (Lin et al., 2024) observes that preserving a small subset of salient weights (identified by activation magnitudes) is crucial for performance, proposing an activation-aware scaling method. Other approaches like QuIP (Chee et al., 2023) apply linear transformation to smooth the weight distribution, ensuring that salient features are preserved even under aggressive quantization. While these methods effectively minimize reconstruction error (e.g., Euclidean distance between full-precision and quantized outputs), they optimize for the frozen pre-trained state in isolation, ignoring the corrective potential available in subsequent fine-tuning. Consequently, the resulting quantized models may possess error structures that are difficult for downstream adapters to correct, limiting their potential in fine-tuning scenarios.

### 2.2. Low Rank Adaptation

To facilitate the adaptation of quantized pre-trained models to downstream tasks efficiently, PEFT methods like LoRA (Hu et al., 2022) have become foundational for adapting large language models (LLMs) by constraining weight updates to low-rank subspaces.

A key research line combines quantization with LoRA-style fine-tuning to balance compression and adaptability. QLoRA (Dettmers et al., 2023) pioneered this direction by introducing the 4-bit NormalFloat (NF4) data type and Double Quantization, enabling the fine-tuning of 65B-parameter models on consumer hardware. Building on this foundation, recent works have sought to refine the interaction between the quantized backbone and the adapter. LQ-LoRA (Guo et al., 2023) decomposes pre-trained matrices into fixed quantized and optimized low-rank components, employing integer linear programming to dynamically assign quantization parameters based on Fisher information and memory budgets. QA-LoRA (Xu et al., 2024) addresses the inference inefficiency by proposing group-wise quantization and adaptation operators, ensuring that the adapter can be perfectly merged into the quantized weights for efficient INT4 inference. Meanwhile, LowRA (Zhou et al., 2025) pushes the boundaries of ultra-low-bit adaptation by optimizing fine-grained quantization hyperparameters, such as mapping and threshold selection, to mitigate the error introduced by aggressive compression.

These works highlight the synergy between low-rank adaptation and quantization, but few explicitly incorporate downstream low-rank fine-tuning constraints during the quantization stage. LoftQ (Li et al., 2023) introduces a LoRA-aware quantization scheme that alternates between weight residual quantization and SVD-based low-rank approximation, alleviating the initialization discrepancy observed in low-bit

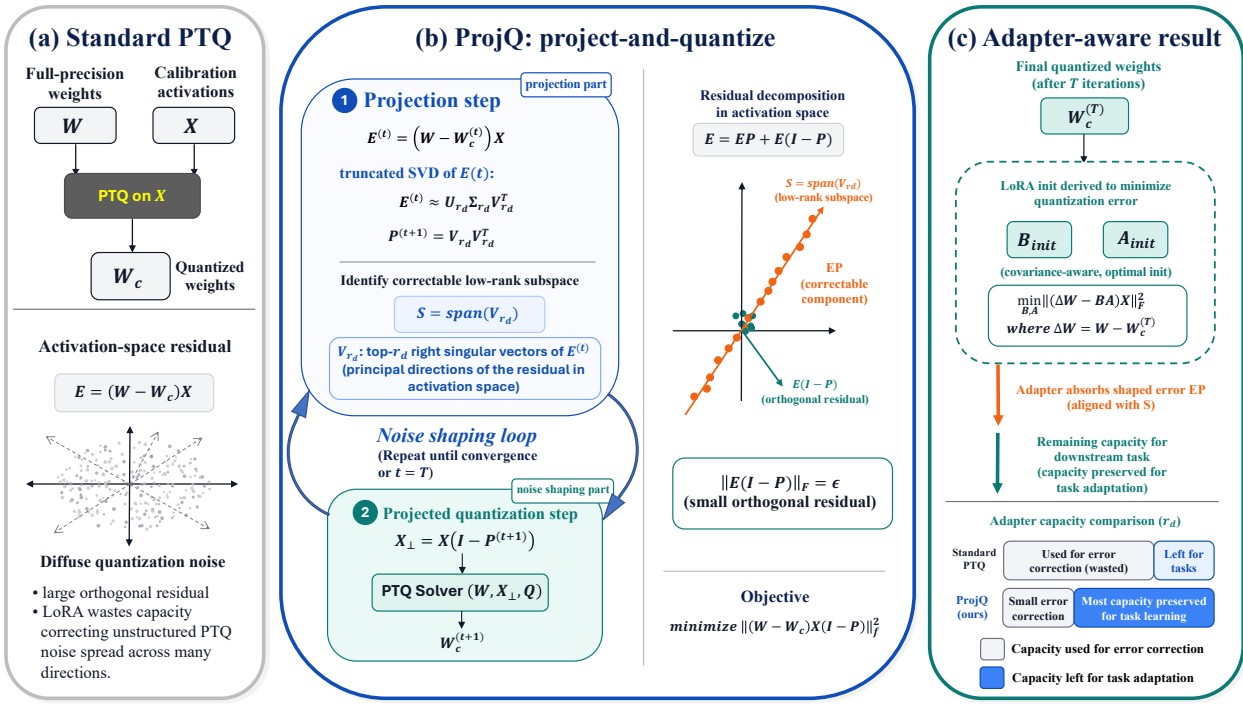

Figure 1. Overview of the proposed framework ProjQ. ProjQ shapes activation-space quantization error into a low-rank subspace that LoRA can correct easily, while minimizing the uncorrectable orthogonal residual.

settings where QLoRA degrades. However, LoftQ operates primarily in the weight space by treating all weights equally, whereas prior PTQ literature suggests that minimizing activation-aware output error is often more effiencient since the contribution of weights to the layer output is considered. To leverage this insight, our approach uses subspace projection to explicitly align quantization with low-rank adaptation, ensuring that both stages jointly target the minimization of output error with theoretical guarantees.

## 3. Problem Formulation

We consider the problem of preparing a quantized LLM layer for effective downstream adaptation. Let $W \in \mathbb{R}^{m \times n}$ be the pre-trained weights and $X \in \mathbb{R}^{n \times N}$ the calibration inputs. Our objective is to generate a quantized backbone $\widehat{W} \in \mathcal{Q}$ (where $\mathcal{Q}$ is a discrete codebook) and initialize a low-rank adapter ($B \in \mathbb{R}^{m \times r}, A \in \mathbb{R}^{r \times n}$) for subsequent finetuning.

Since the downstream task shift is unknown during quantization, we cannot explicitly optimize for the future fine-tuning loss. Instead, we propose to optimize the quantized backbone locally to maximize its *correctability*. More precisely, we would like to design a quantization scheme such that a large portion of the quantization noise is confined to a subspace which could be compensated more easily by the

following low rank adapter. To this end, we minimize the layer-wise reconstruction error *after* applying an optimal low-rank correction:

$$\min_{\widehat{W} \in \mathcal{Q}} \min_{B \in \mathbb{R}^{m \times r}, A \in \mathbb{R}^{r \times n}} \left\| (W - \widehat{W} + BA)X \right\|_F^2. \quad (1)$$

A critical feature of formulation (1) is the explicit inclusion of input activations $X$. Prior works like LoftQ (Li et al., 2023) typically minimize the error in the weight space. By ignoring $X$, such methods treat all weight parameters uniformly, often wasting bit-budget on weights that have minimal impact on the layer's output. In contrast, our objective weighs the quantization error by its actual contribution to the output variance. This ensures that the limited capacity of the adapter is focused on correcting errors in the most salient feature directions, which is essential for preserving performance in low-bit regimes.

Solving the optimization problem in Eq. (1) effectively addresses two distinct goals for efficient fine-tuning: **(i) Structural Noise Shaping.** The outer minimization over $\widehat{W}$ does not simply minimize the magnitude of the noise, but shapes its *structure*. By anticipating the inner correction term $BA$, the quantizer is forced to offload the dominant error components into the low-rank subspace, leaving only the "uncorrectable" orthogonal residual; **(ii) Optimal Adapter**

**Initialization.** The inner minimization yields the specific matrices $(A, B)$ that maximally cancel the local quantization error. While these are not the final task parameters, they provide the mathematically optimal initialization for the adaptation phase to mitigate the degradation induced by quantization, allowing the subsequent fine-tuning to better focus on learning new tasks.

# 4. Methodology: ProjQ

The objective formulated in Eq. (1) presents a formidable optimization challenge. It is a mixed-discrete-continuous optimization problem where the discrete variable $\widehat{W}$ (constrained to the codebook $\mathcal{Q}$) is coupled with the continuous variables $B$ and $A$ inside the Frobenius norm. Solving for all variables simultaneously is computationally intractable.

To address this optimization challenge, we adopt an alternating minimization strategy that decouples the noise-shaping objective from the final adaptation capacity. Our framework distinguishes between two key parameters: the designed rank ($r_d$), which governs the dimensionality of the subspace used to shape the quantization noise during the alternating minimization phase, and the adaptation rank ($r_a$), which determines the actual capacity of the low-rank adapter available for the initialization and subsequent fine-tuning. Accordingly, the approach proceeds in two stages: first, we iteratively optimize the quantized weights and the projector using $r_d$ to confine errors within the target subspace; second, we adopt a covariance-aware decomposition using $r_a$ to derive the optimal adapter initialization. A schematic overview of these techniques is provided in Figure 1.

## 4.1. Equivalence via Subspace Projection

We first establish that optimizing a low-rank adapter of rank $r_d$ is mathematically equivalent to identifying an orthogonal projector $P$ that maximizes the error captured in that subspace. This transformation allows us to optimize the "correctability" of the quantization noise without explicitly solving for adapter parameters at every step.

**Proposition 4.1** (Projection Equivalence). *Let $R(\widehat{W}) = (W - \widehat{W})X$ be the quantization residual in the data space. The minimization over low-rank factors $B \in \mathbb{R}^{m \times r_d}$ and $A \in \mathbb{R}^{r_d \times n}$ is equivalent to minimization over the set of rank-$r_d$ orthogonal projectors $\mathcal{P}_{r_d}$:*

$$\min_{B,A} \left\| R(\widehat{W}) + BAX \right\|_F^2 = \min_{P \in \mathcal{P}_{r_d}} \left\| R(\widehat{W})(I - P) \right\|_F^2,$$

(2)

*where $\mathcal{P}_{r_d} := \{P \in \mathbb{R}^{N \times N} \mid P^2 = P, P^\top = P, \operatorname{rank}(P) = r_d\}$.*

This proposition reveals a fundamental geometric insight: optimizing the low-rank adapter is equivalent to identifying

the subspace where the quantization error is most significant. The term $R(\widehat{W})(I - P)$ represents the residual error projected onto the *orthogonal complement* of the subspace $P$. Therefore, minimizing this residual is equivalent to finding a projector $P$ that captures (and thus "corrects") the maximum possible energy of the quantization error.

## 4.2. Alternating Optimization: project-and-quantize

Substituting Eq. (2) into our original objective yields the *Projector Form* of the problem, governed by the designed rank $r_d$:

$$\min_{\widehat{W} \in \mathcal{Q}} \min_{P \in \mathcal{P}_{r_d}} \left\| (W - \widehat{W})X(I - P) \right\|_F^2.$$

(3)

This reformulation reveals that we can solve for $\widehat{W}$ and the subspace projector $P$ alternatingly.

**-Subspace Identification (P-Step):** Fixing $\widehat{W}$, we identify the optimal projector $P$. Based on Proposition 4.1, $P$ corresponds to the principal subspace of the current residual. We compute $R^{(t)} = (W - \widehat{W}^{(t)})X$ and perform truncated SVD to obtain the top-$r_d$ right singular vectors $V_{r_d}^{(t)}$. The update is:

$$P^{(t+1)} = V_{r_d}^{(t)}(V_{r_d}^{(t)})^\top.$$

(4)

This step identifies the dominant $r_d$ error directions that are most "correctable."

**-Projected Quantization (W-Step):** Fixing $P^{(t+1)}$, we optimize the discrete weights. We project the input activations onto the orthogonal complement of the designed subspace: $X_\perp = X(I - P^{(t+1)})$. The objective becomes:

$$\min_{\widehat{W} \in \mathcal{Q}} \|(W - \widehat{W})X_\perp\|_F^2.$$

(5)

This is structurally identical to a standard activation-aware PTQ problem (e.g., GPTQ), but with $X_\perp$ replacing $X$. Because $X_\perp$ has nullified the directions handled by the designed subspace, the PTQ solver automatically focuses the bit-budget on the remaining "uncorrectable" tail errors.

## 4.3. Optimal Adapter Initialization

After obtaining the quantized weights $\widehat{W}$ via the alternating optimization (using $r_d$), the final step is to initialize the actual low-rank adapter with rank $r_a$ for fine-tuning.

To align the adapter with the actual layer output error, we recall the covariance-aware decomposition strategy introduced in EoRA (Liu et al., 2024) and SVD-LLM (Wang et al., 2025). These works established that minimizing activation-weighted error requires projecting the weight residual into an eigenspace defined by the activation covariance. We adopt this result to derive the optimal initialization for our adaptation rank $r_a$:

**Lemma 4.2** (Covariance-Aware Initialization (Liu et al., 2024)). *Let $\Delta W = W - \widehat{W}$ be the quantization residual and $XX^\top = U_X \Lambda_X U_X^\top$ be the eigendecomposition of the activation covariance. Define the whitening matrix $Y = U_X \Lambda_X^{1/2}$. If $U_{r_a} \Sigma_{r_a} V_{r_a}^\top$ is the rank-$r_a$ truncated SVD of the projected residual $\Delta W Y$, then the optimal adapter factors $(B, A)$ minimizing $\|(\Delta W - BA)X\|_F^2$ are:*

$$B_{init} = U_{r_a} \Sigma_{r_a}^{1/2} \tag{6}$$

$$A_{init} = \Sigma_{r_a}^{1/2} V_{r_a}^\top \Lambda_X^{-1/2} U_X^\top \tag{7}$$

While the solution in Lemma 4.2 involves matrix decompositions, its computational overhead is negligible in the context of offline initialization. The most computationally intensive steps are the eigendecomposition of $XX^\top$ and the SVD of the residual, which are performed only once per layer. The inversion term $\Lambda_X^{-1/2}$ does not require expensive general matrix inversion algorithms (which scale cubically, $O(n^3)$). Since $\Lambda_X$ contains the eigenvalues of the symmetric covariance matrix, it is strictly diagonal. Consequently, computing its inverse reduces to a simple element-wise scalar operation with linear complexity $O(n)$, ensuring both numerical stability and computational efficiency.

This closed-form expression forms the final component of our framework. By combining the subspace-aware quantization (Phase 1) with this covariance-aware initialization (Phase 2), ProjQ ensures that the quantized model is not only structurally optimized to confine errors within a low-rank subspace but also explicitly corrected before fine-tuning begins. This two-stage approach effectively minimizes the "starting penalty" for the downstream task. The complete procedure, integrating the alternating optimization and the closed-form initialization, is summarized in Algorithm 1.

## 5. Theoretical Analysis

In this section, we provide a rigorous theoretical justification for ProjQ. We analyze its robustness to downstream task shifts compared to classical PTQ by deriving an upper bound on the fine-tuning loss.

### 5.1. Theoretical Gains

A core challenge in quantizing Foundation Models is ensuring that the quantized backbone retains sufficient "plasticity" for downstream fine-tuning. To quantify this, we explicitly model the adaptation process as a resource competition between learning a new task and correcting quantization errors.

Let the optimal weights for a specific downstream task be denoted as $W^* = W + S$, where $W$ is the pre-trained weight and $S$ represents the ground truth model shift. Con-

---

**Algorithm 1 ProjQ**: Project-and-Quantize Framework

---
1: **Input:** Weights $W$, Input Activations $X$, Designed Rank $r_d$, Adaptation Rank $r_a$, Codebook $\mathcal{Q}$
2: **Initialize:** $\widehat{W}^{(0)}$ via standard PTQ on $X$
3: **// Phase 1: Alternating Optimization (Noise Shaping)**
4: **for** $t = 0$ **to** $T_{\max}$ **do**
5:    *// P-Step: Identify Correctable Subspace (Rank $r_d$)*
6:    $R^{(t)} \leftarrow (W - \widehat{W}^{(t)})X$
7:    $U, \Sigma, V_{r_d}^\top \leftarrow \text{SVD}_{r_d}(R^{(t)})$
8:    $P^{(t+1)} \leftarrow V_{r_d} V_{r_d}^\top$
9:    *// W-Step: Quantize on Orthogonal Subspace*
10:    $X_\perp \leftarrow X(I - P^{(t+1)})$
11:    $\widehat{W}^{(t+1)} \leftarrow \text{PTQ\_Solver}(W, X_\perp, \mathcal{Q})$
12: **end for**
13: **// Phase 2: Adapter Initialization (Rank $r_a$)**
14: $\Delta W \leftarrow W - \widehat{W}^{(T)}$
15: $U_X, \Lambda_X, \_ \leftarrow \text{SVD}(XX^\top)$   *// Data Covariance*
16: $Y \leftarrow U_X \Lambda_X^{1/2}$
17: $T \leftarrow \Delta W Y$   *// Whitened Residual*
18: $U_{r_a}, \Sigma_{r_a}, V_{r_a}^\top \leftarrow \text{SVD}_{r_a}(T)$   *// SVD with Adaptation Rank*
19: $B_{init} \leftarrow U_{r_a} \Sigma_{r_a}^{1/2}$
20: $A_{init} \leftarrow \Sigma_{r_a}^{1/2} V_{r_a}^\top \Lambda_X^{-1/2} U_X^\top$
21: **Output:** Quantized weights $\widehat{W}^{(T)}$, Adapter $(A_{\text{init}}, B_{\text{init}})$

---

sistent with the hypothesis that adaptation occurs in a low-dimensional subspace (Hu et al., 2022), we assume $S$ has a low intrinsic rank $r_s$. Our goal is to construct a final adapted model $W_{final} = \widehat{W} + L$, where $\widehat{W}$ is the quantized backbone and $L$ is a low-rank adapter of rank $r_a$, such that $W_{final}$ approximates the target $W^*$ as closely as possible.

We define the fine-tuning loss $f(\widehat{W})$ as the minimal activation-weighted error achievable by the adapted model. This formulation reveals that the adapter $L$ must simultaneously fit the task shift $S$ and correct the quantization residual $(W - \widehat{W})$:

$$\begin{aligned} f(\widehat{W}) &= \min_{\text{rank}(L) \le r_a} \|(W_{final} - W^*)X\|_F^2 \\ &= \min_{\text{rank}(L) \le r_a} \left\| (W - \widehat{W})X + SX - LX \right\|_F^2 \end{aligned} \tag{8}$$

To analyze the impact of different quantization strategies on $f(\widehat{W})$, we formally define the quantized weights obtained by Classical PTQ ($W_c$) and ProjQ ($W_p$) as:

$$W_c = \arg\min_{\widehat{W} \in \mathcal{Q}} \|(W - \widehat{W})X\|_F^2 \tag{9}$$

$$W_p = \arg\min_{\widehat{W} \in \mathcal{Q}} \min_{P \in \mathcal{P}_{r_d}} \|(W - \widehat{W})X(I - P)\|_F^2 \tag{10}$$

where $\mathcal{P}_{r_d}$ denotes the set of rank-$r_d$ orthogonal projectors. While $W_c$ minimizes the global error magnitude, $W_p$ minimizes the error projected onto the orthogonal complement of a rank-$r_d$ subspace.

**Proposition 5.1** (Separable Upper Bound). *Consider a feasible adapter $L_{sep}$ constructed by dedicating rank $r_s$ to perfectly fit the task shift $S$, and the remaining capacity $r_a - r_s$ to correcting the quantization error. This yields an upper bound on the fine-tuning loss:*

$$f(\widehat{W}) \leq \mathcal{U}(\widehat{W}) := \mathcal{T}_{r_a - r_s}(E(\widehat{W})) \qquad (11)$$

*where $E(\widehat{W}) = (W - \widehat{W})X$ is the quantization error residual.*

The bound $\mathcal{U}(\widehat{W})$ represents the error remaining after a "safe" adaptation strategy where task learning and error correction are decoupled. A lower $\mathcal{U}$ implies that the quantization noise is structured such that it does not interfere with the task-learning dimensions.

We now establish the dominance of ProjQ under the general condition of a non-zero task shift.

**Theorem 5.2** (Upper Bound Dominance under Task Shift). *Assume the "Abundant Capacity" condition where the adapter rank covers both the designed projection rank and the task shift rank: $r_a \geq r_d + r_s$. Under the assumption that Classical PTQ produces a diffuse error spectrum while ProjQ successfully concentrates error, the separable upper bound of ProjQ is lower than or equal to that of classical PTQ:*

$$\mathcal{U}(W_p) \leq \mathcal{U}(W_c) \qquad (12)$$

Theorem 5.2 demonstrates that ProjQ provides a "cleaner" workspace for the adapter. Even if the downstream task consumes $r_s$ dimensions of the adapter's capacity, the remaining capacity ($r_a - r_s$) is sufficient to correct the ProjQ error because that error has been forced into the null space of the adapter.

While our theoretical analysis utilizes a constructed "separable" adapter to derive the bound, practical fine-tuning (e.g., LoRA) employs a single unified adapter to minimize the joint loss. Theorem 5.2 implies that under ProjQ, this single adapter is not overwhelmed by the "dual burden" of error correction and task learning. Since the quantization error is structurally confined to a low-rank subspace, the optimizer can efficiently allocate a small fraction of the adapter's parameters to cancel this noise, effectively preserving the vast majority of the adapter's capacity to focus on the downstream task shift. This contrasts with standard PTQ, where the adapter must exhaust its limited rank fitting diffuse, high-dimensional noise, leaving insufficient capacity for the actual task.

**Corollary 5.3** (Optimality in Zero-Shift Regime). *Consider the special case where $r_s = 0$ (no task shift, pure reconstruction). If the adapter rank satisfies $r_a \geq r_d$, then:*

$$f(W_p) \leq f(W_c) \qquad (13)$$

Corollary 5.3 confirms that ProjQ acts as a spectral filter. It guarantees that an adapter of sufficient rank can mitigate the quantization errors more effectively than it could for the unstructured noise produced by Classical PTQ.

### 5.2. Convergence Analysis

The ProjQ framework can be implemented as an alternating minimization algorithm, solving for the optimal quantization $\widehat{W}$ and the optimal subspace projector $P$ iteratively. Here, we prove the convergence of this procedure.

Let $\mathcal{J}(\widehat{W}, P)$ be the ProjQ objective function defined in Eq. (9):

$$\mathcal{J}(\widehat{W}, P) = \|(W - \widehat{W})X(I - P)\|_F^2 \qquad (14)$$

where $\widehat{W} \in \mathcal{Q}$ is constrained to the discrete quantization codebook, and $P \in \mathcal{P}_{r_a}$ is a rank-$r_q$ orthogonal projector.

**Proposition 5.4** (Monotonic Convergence). *Let $\{(\widehat{W}^{(t)}, P^{(t)})\}_{t \geq 0}$ be the sequence generated by alternating updates:*

$$P^{(t+1)} = \arg\min_{P \in \mathcal{P}_{r_a}} \mathcal{J}(\widehat{W}^{(t)}, P) \qquad (15)$$

$$\widehat{W}^{(t+1)} = \arg\min_{\widehat{W} \in \mathcal{Q}} \mathcal{J}(\widehat{W}, P^{(t+1)}) \qquad (16)$$

*Assuming the inner PTQ solver used for Eq. (16) ensures non-increasing error, the sequence of objective values $\{\mathcal{J}_t\}_{t \geq 0}$ is monotonically non-increasing and converges to a stationary value $\mathcal{J}^*$.*

This analysis ensures that ProjQ is numerically stable. In practice, the proposed algorithm stabilizes rapidly, often requiring only 1-3 iterations.

## 6. Experiments

In this section, we empirically validate the proposed ProjQ framework. Our evaluation is designed to probe two distinct capabilities of the method: (1) the ability to compensate for quantization error in a zero-shot setting, effectively treating the adapter as a restoration mechanism; and (2) the ability to facilitate downstream adaptation, where the adapter must simultaneously correct quantization errors and learn new task-specific features.

### 6.1. Experimental Setup

**Models and Datasets.** To ensure a comprehensive evaluation, we conduct experiments on three representative families of Large Language Models: LLaMA2 (Touvron et al.,

*Table 1.* Results on general NLU benchmarks for LLaMA2, Qwen2.5 and Qwen3 models under 2-bit quantization ($r_a = r_d = 64$).

| Metric | Method | LLaMA2 | | Qwen2.5-Ins | | | Qwen3 | | |
|---|---|---|---|---|---|---|---|---|---|
| | | 7B | 13B | 7B | 14B | 32B | 4B | 8B | 32B |
| **C4 (PPL, ↓)** | GPTQ+SVD-LLM | 26.26 | 14.50 | 62.27 | 28.94 | 16.96 | 88.43 | 52.65 | 26.24 |
| | AWQ+SVD-LLM | 1.7e+5 | 9.5e+4 | NAN | NAN | – | – | – | – |
| | CALDERA | 21.59 | 13.56 | 50.57 | 23.33 | – | 96.90 | **39.36** | – |
| | LoftQ | 28.77 | 14.14 | 34.17 | 31.74 | 16.95 | 105.34 | 54.71 | 25.71 |
| | **ProjQ** | **21.50** | **12.48** | **33.50** | **22.22** | **14.33** | **83.64** | 42.02 | **20.74** |
| **WikiText (PPL,↓)** | GPTQ+SVD-LLM | 28.81 | 13.76 | 92.21 | 27.51 | 15.30 | 116.56 | 76.19 | 33.69 |
| | AWQ+SVD-LLM | 2.2e+5 | 1.2e+5 | NAN | NAN | – | – | – | – |
| | CALDERA | 23.82 | 12.99 | 70.52 | 22.78 | – | 116.61 | 49.36 | – |
| | LoftQ | 30.02 | 12.80 | 40.36 | 28.10 | 15.46 | 140.70 | 62.38 | 33.01 |
| | **ProjQ** | **22.42** | **11.56** | **38.66** | **21.50** | **12.45** | **101.22** | **48.75** | **21.18** |
| **Common Sense Avg.ACC (↑, %)** | GPTQ+SVD-LLM | 45.33 | 52.97 | 40.96 | 46.28 | 49.77 | 38.38 | 39.55 | 44.09 |
| | AWQ+SVD-LLM | 36.61 | 37.82 | 37.51 | 36.67 | – | – | – | – |
| | CALDERA | 46.85 | 54.02 | 42.50 | **47.78** | – | 38.90 | 41.15 | – |
| | LoftQ | 45.19 | 53.82 | 43.90 | 46.10 | 53.42 | 38.81 | 40.98 | 44.88 |
| | **ProjQ** | **47.82** | **55.58** | **44.44** | 47.68 | **55.05** | **39.28** | **41.96** | **47.23** |

2023), Qwen2.5-Instruct and Qwen3(Bai et al., 2023). We evaluate the models across a diverse set of benchmarks covering language modeling, commonsense reasoning, and mathematical reasoning. For language modeling, we report perplexity (PPL) on the WikiText-2 (Merity et al., 2016), PTB (Marcus et al., 1994), and C4 (Raffel et al., 2020) test datasets. For commonsense reasoning, we report the average zero-shot accuracy on four widely used datasets: ARC-Challenge (ARC-C), ARC-Easy (ARC-E) (Clark et al., 2018), PIQA (Bisk et al., 2020), and StoryCloze (Mostafazadeh et al., 2016). Mathematical reasoning capabilities are assessed using the GSM8K (Cobbe et al., 2021) benchmark.

**Baselines and Implementation Details.** We compare ProjQ with representative PTQ, LoRA-aware quantization, and low-rank compensation baselines, including **GPTQ** (Frantar et al., 2022), **AWQ** (Lin et al., 2024), **LoftQ** (Li et al., 2023), **QLoRA** (Dettmers et al., 2023), **SVD-LLM** (Wang et al., 2025), and **CALDERA** (Saha et al., 2024). For fair comparison, GPTQ and AWQ are combined with the same SVD-LLM-style adapter construction, while LoftQ and CALDERA are evaluated as LoRA-aware low-rank quantization baselines. For QLoRA, we use GPTQ-initialized backbones on WikiText-2 and GSM8K, since RTN is unstable in the 2-bit and 3-bit regimes (Li et al., 2023); on commonsense reasoning tasks, we report both RTN- and GPTQ-initialized variants.

All experiments are implemented in PyTorch using the HuggingFace Transformers library. We conduct quantization experiments under 2-bit with a group size of 128. Consistent with standard PTQ protocols, we utilize a calibration set consisting of 128 random sequences of length 2048 sampled from the C4 dataset. The PTQ solver used in ProjQ

is GPTQ. The maximum number of iterations is set as 5. All evaluations are performed on a single NVIDIA A100 (80GB) GPU.

### 6.2. Performance comparison

**Quantization Error Compensation.** We first isolate the efficacy of ProjQ in the regime of pure error compensation. In this setting, the low-rank adapter is not trained on a downstream task but is instead initialized to minimize the reconstruction error of the quantized backbone. This corresponds to the theoretical case of zero task shift discussed in Corollary 5.3, where the objective is strictly to recover the pre-trained capabilities. The adapter for all three schemes are the one given by Lemma 4.2, which has proven to be the best low rank matrices to compensate the quantization error. The results for LLaMA2, Qwen2.5 and Qwen3 under aggressive 2-bit quantization are summarized in Table 1.

Table 1 shows that ProjQ provides consistently stronger error compensation than the baselines under 2-bit quantization. Across LLaMA2, Qwen2.5-Instruct, and Qwen3 models, ProjQ achieves the lowest perplexity in most language modeling evaluations and competitive or higher commonsense reasoning accuracy. For example, on LLaMA2-7B, ProjQ reduces the C4 perplexity from 26.26 with GPTQ+SVD-LLM and 28.77 with LoftQ to 21.50, while also improving the commonsense average accuracy from 45.33% and 45.19% to 47.82%. These improvements support the motivation of ProjQ: for low-rank compensation, not only the magnitude but also the structure of the quantization error matters. Standard PTQ-based methods mainly reduce the overall reconstruction error, but the remaining error can still be spread across many activation-space directions, making it difficult for a rank-limited adapter to correct. In contrast,

*Table 2.* Fine-tuning results across Common Sense Reasoning, WikiText-2, and GSM8K tasks.

| Task | Bit | Method | LLaMA2-7B | Qwen2.5-7B-Ins | LLaMA2-13B | Qwen2.5-14B-Ins |
|---|---|---|---|---|---|---|
| | 16 | FP16 | 68.07 | 69.84 | 70.61 | 73.44 |
| | 2 | QLoRA(GPTQ) | 57.06 | 59.66 | - | - |
| | | QLoRA(RTN) | 54.64 | 56.64 | - | - |
| | | LoftQ | 54.78 | 58.86 | 60.25 | 59.08 |
| Common Sense(Avg.↑, %) | | **ProjQ** | **57.59** | **59.88** | **61.84** | **59.85** |
| | 3 | QLoRA(GPTQ) | 66.15 | 65.08 | - | - |
| | | QLoRA(RTN) | 66.56 | 64.91 | - | - |
| | | LoftQ | 66.38 | 64.67 | 69.10 | 69.89 |
| | | **ProjQ** | **67.13** | **68.33** | **69.91** | **71.61** |
| | 16 | FP16 | 5.18 | 7.16 | 4.66 | 5.97 |
| | 2 | QLoRA | 8.73 | 13.13 | - | - |
| | | LoftQ | 9.14 | 13.15 | 7.33 | 12.17 |
| WikiText-2 (PPL ↓) | | **ProjQ** | **8.17** | **12.85** | **6.44** | **12.01** |
| | 3 | QLoRA | 6.13 | 8.93 | - | - |
| | | LoftQ | 6.20 | 8.82 | 5.09 | 7.31 |
| | | **ProjQ** | **5.69** | **8.02** | **4.98** | **7.08** |
| | 16 | FP16 | 38.42 | 70.89 | 46.63 | 82.52 |
| | 2 | QLoRA | 22.29 | 44.73 | - | - |
| | | LoftQ | 22.52 | 46.85 | 30.98 | **51.25** |
| GSM8K(Avg.↑,%) | | **ProjQ** | **22.90** | **47.16** | **31.25** | 51.11 |
| | 3 | QLoRA | 36.03 | 69.52 | - | - |
| | | LoftQ | **37.68** | 70.02 | 45.03 | 78.39 |
| | | **ProjQ** | 35.18 | **70.81** | **46.07** | **81.50** |

ProjQ explicitly optimizes the quantized backbone so that the dominant quantization error is concentrated in a low-rank, adapter-correctable subspace, while the residual error in the orthogonal complement is minimized. Therefore, the covariance-aware adapter initialization in Lemma 4.2 can more effectively cancel the shaped error. Additional results with different bit-widths and rank settings are reported in Appendix B, which further confirm the robustness of this error-shaping effect.

**Fine-tuning for Downstream Tasks.** We next evaluate the fine-tuning pipeline, where the initialized adapter is trained on downstream datasets. This scenario introduces the dual burden described in Section 5.1, requiring the adapter to allocate capacity for both error correction and task learning ($r_s > 0$). The results are shown in Table 2.

For commonsense reasoning, we fine-tune the quantized models on the Commonsense-170k dataset (Talmor et al., 2019) and evaluate average accuracy across four benchmarks. ProjQ consistently enables higher downstream accuracy compared to LoftQ and QLoRA. Notably, under the challenging 2-bit quantization setting for Llama2-7B, ProjQ achieves an average accuracy of 57.59%, surpassing the LoftQ baseline of 54.78%. This trend persists across 3-bit

and 4-bit settings, where ProjQ maintains a robust lead. In the domain of language modeling, fine-tuning on WikiText-2 reveals that ProjQ yields consistently lower perplexity. For LLaMA-2-7B quantized to 2 bits, ProjQ reaches a perplexity of 8.17, while LoftQ saturates at 9.14. Moreover, the 3-bit ProjQ model achieves a perplexity of 5.69, reaching the same level of performance of the 4-bit LoftQ and QLoRA baseline (see Appendix B), effectively enabling a 25% reduction in model size without compromising generation quality. Finally, on the GSM8K mathematical reasoning benchmark, ProjQ achieves slightly better accuracy.

These results provide strong empirical support for the theoretical analysis. By offloading the uncorrectable quantization error into a subspace that is pre-compensated by the initialization, ProjQ effectively preserves the plasticity of the adapter. Unlike standard approaches where the optimizer must waste capacity suppressing unstructured noise, ProjQ provides a cleaner initialization state, allowing the limited rank of the adapter to be fully utilized for learning complex downstream task features.

*Table 3.* Error compensation results of ProjQ with different iterations on LLaMA2-7B.

| | PPL ($\downarrow$) | | | ACC ($\uparrow$, %) |
|---|---|---|---|---|
| Iteration | C4 | PTB | WikiText | Avg. |
| 1 | 25.62 | 634.14 | 30.75 | 46.91 |
| 2 | 25.25 | 2.5e+3 | 29.55 | 40.91 |
| 3 | 25.32 | 644.42 | 28.99 | 46.00 |

### 6.3. Ablation study

**Impact of Alternating Iterations.** We investigate the convergence behavior of the proposed algorithm by varying the number of alternating iterations ($T_{max}$). Table 3 presents the quantization error compensation results on LLaMA2-7B (2-bit) for the first three iterations. We observe that ProjQ converges rapidly. A single iteration is sufficient to identify a high-quality subspace. Extending the optimization to 3 iterations further refines the alignment. Although we utilized a conservative setting of $T_{max} = 5$ for our main experiments to ensure stability, these results demonstrate that ProjQ can achieve competitive performance with as few as 1-3 iterations, making it highly computationally efficient for offline quantization.

**Comparison of Initialization Scheme.** To evaluate the effectiveness of the adapter initialization strategy proposed in this work, we conduct an ablation study under identical experimental settings. Specifically, based on the 2-bit quantized model obtained in Phase 1 of ProjQ, We compare LoRA fine-tuning results with standard LoRA initialization against our proposed initialization in Table 4. Our proposed initialization consistently outperforms the standard random LoRA initialization across both models and evaluation metrics. These improvements indicate that the proposed initialization yields a better optimization starting point, thereby enhancing the effectiveness of Phase 2 adaptation. More ablation studies can be found in Appendix B.

*Table 4.* Error compensation results of ProjQ with different LoRA adapter initialization scheme.

| | LLaMA-2-7B | | Qwen2.5-7B-Ins | |
|---|---|---|---|---|
| Initialization | PPL($\downarrow$) | Avg.($\uparrow$) | PPL($\downarrow$) | Avg.($\uparrow$) |
| Standard LoRA | 9.17 | 56.69 | 16.40 | 53.66 |
| **Proposed Init** | **8.17** | **57.59** | **12.85** | **59.88** |

### 6.4. Scalability, computational complexity, and time overhead

Regarding computational overhead, it has to be noticed that there is low risk of SVD or covariance operations bottlenecking larger models. Because ProjQ projects noise onto a tightly constrained subspace ($r_d \ll d$), we utilize truncated randomized SVD to extract only the top-$r_d$ components instead of computing the full SVD ($\mathcal{O}(d^3)$). This reduces time complexity to $\mathcal{O}(d^2 \cdot r_d)$, yielding an approximate $30\times$ to $100\times$ theoretical reduction in FLOPs compared to full SVD for a standard layer. During initialization, ProjQ's computational complexity is highly comparable to LoftQ. While LoftQ relies on the SVD of the residual weight matrix ($W - \widehat{W}$), our covariance-aware initialization (Lemma 4.2) achieves similar theoretical efficiency. Activation-based SVD typically requires dense matrix inversions, but our formulation strictly inverts a diagonal matrix, avoiding standard computational bottlenecks. Consequently, the empirical wall-clock times for initializing LLaMA2-13B are highly comparable (**13 min for LoftQ vs. 17 min for ProjQ**). This minor difference is brought entirely by the necessary computation of the activation covariance matrices, which is computationally inexpensive and does not pose a significant delay.

## 7. Conclusion

In this work, we have presented ProjQ, a framework that fundamentally rethinks the interaction between post-training quantization and low-rank adaptation. Rather than treating these stages in isolation, we introduced the "project-and-quantize" paradigm, which actively shapes quantization noise into a low-rank structure that is mathematically aligned with the adapter's corrective capacity. Our theoretical analysis and empirical results converge on a singular insight: in the context of fine-tunable models, the *structure* of the quantization error is as critical as its magnitude. By forcing the dominant error components into a subspace that the adapter is initialized to cancel, ProjQ effectively mitigates the "adaptation burden" from the task learning objective. This ensures that even under aggressive compression, the quantized backbone retains sufficient plasticity to learn complex downstream tasks, a property we have validated across diverse language and reasoning benchmarks. While the current instantiation of ProjQ demonstrates significant gains using standard fixed-bitwidth uniform quantization solvers (e.g., GPTQ), the framework is inherently solver-agnostic and extensible, and both perplexity and accuracy may need to be further improved. A promising avenue for future research is to integrate more sophisticated quantization schemes into the projection loop. For instance, non-uniform quantizers or mixed-precision strategies, such as those explored in LowRA (Zhou et al., 2025), can be implemented within our alternating minimization (W-step). By dynamically allocating bit-budgets based on the spectral "correctability" of specific features, such extensions could further minimize the residual error in the orthogonal uncorrectable subspace, potentially unlocking high-fidelity performance at even lower bit-rates.

## Acknowledgements

We thank the reviewers for their useful comments to improve the quality of this work. This work is partially supported by the Natural Science Foundation of China under Grant No. 62401635.

## Impact Statement

This paper presents work whose goal is to advance the field of Machine Learning. There are many potential societal consequences of our work, none which we feel must be specifically highlighted here.

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

# A. Proofs

## A.1. Proof of Proposition 4.1

*Proof.* Let $R \equiv R(\widehat{W})$ for brevity. We analyze the inner minimization term $\min_{B,A} \|R + BAX\|_F^2$. The product $Z = BAX$ represents the correction applied by the adapter in the output space. This term is constrained by $\mathrm{rank}(Z) \leq r$ and $\mathrm{row}(Z) \subseteq \mathrm{row}(X)$. Since $R = (W - \widehat{W})X$, its rows are linear combinations of $X$, satisfying $\mathrm{row}(R) \subseteq \mathrm{row}(X)$.

By the Eckart-Young-Mirsky theorem, the unconstrained rank-$r$ matrix that minimizes $\|R + Z\|_F^2$ is given by the truncated SVD of $-R$. Let the SVD of the residual be $R = U\Sigma V^\top$. The optimal correction is $Z^* = -U_{r_d}\Sigma_{r_d}V_{r_d}^\top$, where $U_{r_d}, \Sigma_{r_d}, V_{r_d}$ correspond to the top-$r_d$ singular components. Since $\mathrm{row}(Z^*) = \mathrm{span}(V_r^\top) \subseteq \mathrm{row}(R) \subseteq \mathrm{row}(X)$, $Z^*$ is a feasible solution.

We can express this optimal correction using an orthogonal projector. Let $P^* = V_{r_d}V_{r_d}^\top$. Note that $P^*$ is symmetric, idempotent ($P^*P^* = P^*$), and has rank $r_d$, so $P^* \in \mathcal{P}_{r_d}$. We can rewrite the optimal correction as $Z^* = -RP^*$. Since the optimal solution corresponds to an orthogonal projection onto the principal subspace of the error, restricting the search space to orthogonal projectors yields the equivalent minimum:

$$\min_{B,A} \|R + BAX\|_F^2 = \|R - RP^*\|_F^2 = \min_{P \in \mathcal{P}_{r_d}} \|R(I - P)\|_F^2. \tag{17}$$

$\square$

## A.2. Proof of Lemma 4.2

*Proof.* The objective function is the activation-weighted reconstruction error:

$$\mathcal{L} = \|(\Delta W - BA)X\|_F^2. \tag{18}$$

Using the properties of the Frobenius norm and the trace operator ($\|M\|_F^2 = \mathrm{Tr}(MM^\top)$), we rewrite the loss in terms of the data covariance $XX^\top$:

$$\mathcal{L} = \mathrm{Tr}\left((\Delta W - BA)XX^\top(\Delta W - BA)^\top\right). \tag{19}$$

Substituting the eigendecomposition $XX^\top = U_X\Lambda_X U_X^\top$, we define the symmetric "whitening" matrix $Y = U_X\sqrt{\Lambda_X}$, such that $YY^\top = XX^\top$. This allows us to factorize the covariance term inside the trace:

$$\mathcal{L} = \mathrm{Tr}\left((\Delta W - BA)YY^\top(\Delta W - BA)^\top\right) = \|(\Delta W - BA)Y\|_F^2. \tag{20}$$

We now define a transformed adapter $\widetilde{A} = AY$. The optimization problem becomes finding the optimal low-rank product $Z = B\widetilde{A}$ to approximate the target matrix $T = \Delta WY$:

$$\min_{B,\widetilde{A}} \|\Delta WY - B\widetilde{A}\|_F^2. \tag{21}$$

By the Eckart-Young-Mirsky theorem, the optimal rank-$r_a$ matrix $Z^*$ is given by the truncated SVD of the target $T$. Let the rank-$r_a$ SVD of the projected residual be $\Delta WY \approx U_{r_a}\Sigma_{r_a}V_{r_a}^\top$. We decompose the optimal $Z^*$ symmetrically into $B$ and $\widetilde{A}$:

$$B^* = U_{r_a}\Sigma_{r_a}^{\frac{1}{2}}, \quad \widetilde{A}^* = \Sigma_{r_a}^{\frac{1}{2}}V_{r_a}^\top. \tag{22}$$

Finally, we recover the original adapter $A^*$ by inverting the transformation $\widetilde{A}^* = A^*Y$. Since $Y$ is invertible (assuming sufficient calibration data for full rank covariance), we have:

$$A^* = \widetilde{A}^*Y^{-1} = (\Sigma_{r_a}^{\frac{1}{2}}V_{r_a}^\top)(U_X\Lambda_X)^{-\frac{1}{2}} = \Sigma_{r_a}^{\frac{1}{2}}V_{r_a}^\top\Lambda_X^{-\frac{1}{2}}U_X^\top. \tag{23}$$

This completes the proof. $\square$

## A.3. Proof of Proposition 5.1

*Proof.* The fine-tuning loss is defined as the minimal error achievable by an adapter $L$ of rank at most $r_a$:

$$f(\widehat{W}) = \min_{\mathrm{rank}(L) \leq r_a} \|(W - \widehat{W})X + SX - L\|_F^2 \tag{24}$$

Let $E = (W - \widehat{W})X$ denote the quantization error in the activation space. The objective becomes minimizing $\|E + SX - L\|_F^2$.

To derive an upper bound, we construct a specific candidate adapter $L_{sep}$ that is feasible (satisfies the rank constraint) but not necessarily optimal. We construct $L_{sep}$ by explicitly separating the capacity into two components:

1. **Task Component ($L_S$):** We allocate $r_s$ rank to perfectly match the task shift. Let $L_S = SX$. Since $\text{rank}(S) = r_s$, we have $\text{rank}(L_S) \leq r_s$.

2. **Error Component ($L_E$):** We allocate the remaining capacity $k = r_a - r_s$ to approximate the quantization error $E$. By the Eckart-Young-Mirsky theorem, the optimal rank-$k$ approximation of $E$ is given by its truncated SVD. Let $L_E = \text{SVD}_k(E)$. The approximation error is the sum of the squared singular values of the tail: $\|E - L_E\|_F^2 = \sum_{j=k+1}^{N} \sigma_j^2(E) = \mathcal{T}_k(E)$.

We define the separable adapter as $L_{sep} = L_S + L_E$. By the subadditivity of rank, $\text{rank}(L_{sep}) \leq \text{rank}(L_S) + \text{rank}(L_E) \leq r_s + (r_a - r_s) = r_a$. Thus, $L_{sep}$ is a valid candidate for the minimization problem.

Evaluating the loss with this specific candidate yields:

$$\|E + SX - L_{sep}\|_F^2 = \|E + SX - (SX + L_E)\|_F^2 = \|E - L_E\|_F^2 = \mathcal{T}_{r_a - r_s}(E) \tag{25}$$

Since $f(\widehat{W})$ is the infimum over all valid adapters, it must be less than or equal to the error produced by any specific candidate:

$$f(\widehat{W}) \leq \|E + SX - L_{sep}\|_F^2 = \mathcal{T}_{r_a - r_s}(E(\widehat{W})) \tag{26}$$

$\square$

## A.4. Proof of Theorem 5.2

*Proof.* Let $k = r_a - r_s$. The condition $r_a \geq r_d + r_s$ implies $k \geq r_d$. The upper bound is defined as the tail energy starting from index $k$: $\mathcal{U}(\widehat{W}) = \sum_{j=k+1}^{N} \sigma_j^2(E(\widehat{W}))$.

1. **Analysis of $W_p$:** The objective function of ProjQ in Eq. (10) is equivalent to minimizing the projected residual. By the Eckart-Young-Mirsky theorem, minimizing $\|E(I - P)\|_F^2$ over rank-$r_d$ projectors is equivalent to minimizing the tail energy starting at $r_d$:

$$W_p = \arg\min_{\widehat{W}} \sum_{j=r_d+1}^{N} \sigma_j^2(E(\widehat{W})) \tag{27}$$

Since $k \geq r_d$, the summation domain for $\mathcal{U}(W_p)$ (indices $k + 1 \rightarrow N$) is a subset of the domain minimized by ProjQ (indices $r_d + 1 \rightarrow N$). ProjQ explicitly suppresses these singular values.

2. **Analysis of $W_c$:** Classical PTQ minimizes the total Frobenius norm $\|E\|_F^2 = \sum_{j=1}^{N} \sigma_j^2(E)$. Without the subspace constraint, standard quantization algorithms distribute error effectively uniformly across all dimensions to minimize the total sum, resulting in a flat spectrum where $\sigma_j \approx \epsilon$.

3. **Comparison:** Because $W_p$ concentrates the error energy into the top $r_d$ components (which are then discarded by the sum starting at $k \geq r_d$), the tail energy of $W_p$ decays rapidly. Conversely, $W_c$ maintains significant energy in the tail due to its diffuse spectrum. Thus:

$$\sum_{j=k+1}^{N} \sigma_j^2(E(W_p)) \leq \sum_{j=k+1}^{N} \sigma_j^2(E(W_c)) \implies \mathcal{U}(W_p) \leq \mathcal{U}(W_c). \tag{28}$$

$\square$

## A.5. Proof of Corollary 5.3

*Proof.* When $r_s = 0$, the fine-tuning loss simplifies to the tail energy function: $f(\widehat{W}) = \mathcal{T}_{r_a}(E(\widehat{W}))$. Since $r_a \geq r_d$, the objective of ProjQ directly minimizes a superset of the terms constituting $f(W_p)$. Specifically, ProjQ minimizes $\sum_{j=r_d+1}^{N} \sigma_j^2$, while the loss measures $\sum_{j=r_a+1}^{N} \sigma_j^2$. By the same spectral concentration argument as Theorem 5.3, the tail energy of the structured error $E(W_p)$ is lower than or equal to that of the unstructured error $E(W_c)$ in the region $j > r_a$. $\square$

## A.6. Proof of Proposition 5.4

*Proof.* The proof follows from the block-coordinate descent property of the alternating updates:

1. **P-Step (Subspace Alignment):** For a fixed $\widehat{W}^{(t)}$, the problem reduces to finding the optimal rank-$r_a$ subspace to minimize the residual energy of $R^{(t)} = (W - \widehat{W}^{(t)})X$. By the Eckart-Young-Mirsky theorem, the global minimizer $P^{(t+1)}$ is the projector onto the top-$r_q$ right singular vectors of $R^{(t)}$. This step guarantees $\mathcal{J}(\widehat{W}^{(t)}, P^{(t+1)}) \leq \mathcal{J}(\widehat{W}^{(t)}, P^{(t)})$.

2. **W-Step (Projected Quantization):** For a fixed $P^{(t+1)}$, the problem reduces to a standard PTQ problem on the projected activations $X_\perp = X(I - P^{(t+1)})$. Provided the chosen PTQ solver (e.g., GPTQ or greedy coordinate descent) does not increase the error, we have $\mathcal{J}(\widehat{W}^{(t+1)}, P^{(t+1)}) \leq \mathcal{J}(\widehat{W}^{(t)}, P^{(t+1)})$.

Combining these steps yields $\mathcal{J}_{t+1} \leq \mathcal{J}_t$. Since the Frobenius norm is bounded below by 0, the sequence must converge. $\qquad\square$

# B. Additional Results

**Additional Error Compensation Results.** Table 5 presents the error compensation results on LLaMA2-7B and Qwen2.5-7B-Instruct under different rank settings. We observe that the performance gap between ProjQ and baselines grows as the adapter rank decreases, with ProjQ maintaining high fidelity even at minimal ranks where standard methods degrade. This can be explained by the fact that the vast majority of quantization error energy into a compact low-rank subspace with ProjQ, allowing a small-rank adapter to efficiently capture and cancel dominant noise components that remain diffuse and uncorrectable in isotropic quantization schemes.

*Table 5.* Results on general NLU benchmarks of LLaMA2 and Qwen2.5 model under 2 bit quantization.

| Model | Method | Designed Rank | Adapt. Rank | Perplexity (↓) | | | Accuracy (↑, %) | | | | |
|---|---|---|---|---|---|---|---|---|---|---|---|
| | | | | C4 | PTB | WikiText | Arc-E | Arc-C | PIQA | StoryCloze | Avg. |
| LLaMA2-7B | GPTQ | – | 4 | 33.79 | 927.02 | 38.51 | 33.16 | **28.76** | 59.36 | **61.04** | **45.58** |
| | | – | 16 | **31.29** | **765.62** | **34.87** | 35.09 | **28.76** | 60.12 | **61.84** | **46.45** |
| | | – | 64 | 25.58 | 802.52 | 28.38 | 37.54 | 22.41 | 60.88 | **63.17** | 46.00 |
| | LoftQ | 4 | 4 | 57.90 | 2.2e+4 | 73.26 | 30.53 | 24.75 | 56.91 | 54.30 | 41.62 |
| | | 16 | 16 | 44.95 | 2.5e+4 | 52.15 | 32.63 | 19.40 | 55.98 | 56.65 | 41.16 |
| | | 64 | 64 | 28.77 | 2.1e+4 | 30.02 | 37.72 | 21.74 | 59.90 | 61.41 | 45.19 |
| | ProjQ | 4 | 4 | **27.47** | **307.64** | **31.88** | 37.19 | 23.08 | **60.45** | 59.91 | 45.16 |
| | | 16 | 16 | 31.73 | 835.59 | 41.17 | **36.49** | 20.74 | 57.18 | 59.81 | 43.56 |
| | | 64 | 64 | **21.50** | **347.99** | **22.42** | 40.35 | 25.75 | 62.57 | 62.59 | **47.82** |
| Qwen2.5-7B-Instruct | GPTQ | – | 4 | 117.63 | 362.31 | 202.35 | 27.19 | 21.07 | 54.52 | 51.31 | 38.52 |
| | | – | 16 | 100.45 | 327.68 | 178.18 | 30.53 | 20.40 | 55.98 | 50.26 | 39.29 |
| | | – | 64 | 61.68 | 155.73 | 89.56 | 32.11 | 22.74 | 57.24 | 54.20 | 41.57 |
| | LoftQ | 4 | 4 | 122.67 | 171.50 | 137.00 | 32.98 | 22.41 | **57.78** | 52.86 | 41.51 |
| | | 16 | 16 | 76.22 | 142.97 | 85.16 | **35.96** | 19.73 | 56.15 | 54.20 | 41.51 |
| | | 64 | 64 | 34.17 | 69.00 | 40.36 | 34.39 | 23.75 | **60.55** | 56.92 | 43.90 |
| | ProjQ | 4 | 4 | **52.34** | **115.80** | **63.09** | 37.37 | 24.41 | 55.01 | **56.12** | **43.23** |
| | | 16 | 16 | **45.68** | **97.82** | **61.15** | 34.39 | 21.07 | 57.94 | 54.41 | 41.95 |
| | | 64 | 64 | **33.50** | **65.72** | **38.66** | 36.49 | 23.75 | 60.28 | 57.24 | 44.44 |

**Additional Fine-tuning Results.** Table 6 reports fine-tuning results across Common Sense Reasoning, WikiText-2, and GSM8K tasks under 4-bit quantization on LLaMA2-7B and Qwen2.5-7B-Instruct. Table 7 presents complete results of Common Sense tasks with accuracy in each datasets.

**Additional ablations study: Effects of Rank Decoupling.** In our experiments, we primarily report results under the setting where the designed rank ($r_d$) is equal to the adapter rank($r_a$). To justify this design choice, we conduct an ablation study in which $r_a$ is fixed while varying $r_d$ to evaluate its impact on error compensation performance. As shown in table 8, matching $r_d$ to $r_a$ yields superior performance in most cases. This trend aligns with the fundamental trade-off between the structured noise reduction in the backbone and the error-correction capacity of the adapter.

*Table 6.* Fine-tuning results across Common Sense Reasoning, WikiText-2, and GSM8K tasks under 4-bit quantization.

| Task | Bit | Method | LLaMA2-7B | Qwen2.5-7B-Ins |
|---|---|---|---|---|
| | 16 | FP16 | 68.07 | 69.84 |
| Common Sense(Avg.↑, %) | | QLoRA | 68.63 | **71.59** |
| | 4 | LoftQ | 67.21 | 64.56 |
| | | **ProjQ** | **68.70** | 66.78 |
| | 16 | FP16 | 5.18 | 7.16 |
| WikiText-2 (PPL ↓) | | QLoRA | 5.70 | - |
| | 4 | LoftQ | 5.78 | 8.21 |
| | | **ProjQ** | **5.29** | **7.36** |
| | 16 | FP16 | 38.42 | 70.89 |
| GSM8K(Avg.↑,%) | | QLoRA | 38.2 | - |
| | 4 | LoftQ | 39.35 | 72.02 |
| | | **ProjQ** | **39.42** | **72.10** |

*Table 7.* Complete results of Common Sense tasks with accuracy in each datasets.

| Model | Method | Bit | Arc-E | Arc-C | PIQA | StoryCloze | Avg. |
|---|---|---|---|---|---|---|---|
| | QLoRA(GPTQ) | | 57.89 | **32.44** | 67.36 | **70.55** | 57.06 |
| | QLoRA(RTN) | 2 | 56.67 | 27.42 | 66.16 | 68.31 | 54.64 |
| | LoftQ | | 54.91 | 29.77 | 65.61 | 68.84 | 54.78 |
| | ProjQ | | **61.75** | 30.43 | **68.66** | 69.54 | **57.59** |
| | QLoRA(GPTQ) | | 69.12 | 40.47 | **77.04** | **77.98** | 66.15 |
| LLaMA2-7B | QLoRA(RTN) | 3 | 71.05 | **42.47** | 76.17 | 76.54 | 66.56 |
| | LoftQ | | 71.23 | 40.13 | 76.71 | 77.45 | 66.38 |
| | ProjQ | | **73.86** | 42.14 | 76.77 | 75.73 | **67.13** |
| | QLoRA | | 75.44 | 41.47 | **78.13** | **79.48** | 68.63 |
| | LoftQ | 4 | 71.58 | 42.14 | 77.64 | 77.50 | 67.21 |
| | ProjQ | | **75.79** | **44.82** | 77.42 | 76.75 | **68.70** |
| | QLoRA(GPTQ) | | 62.98 | **37.46** | 67.95 | **70.23** | 59.66 |
| | QLoRA(RTN) | 2 | 61.58 | 30.77 | 67.03 | 67.18 | 56.64 |
| | LoftQ | | 65.26 | 32.44 | **68.72** | 69.00 | 58.86 |
| | ProjQ | | **66.67** | 36.79 | 68.17 | 67.88 | **59.88** |
| | QLoRA(GPTQ) | | 69.12 | 36.45 | 76.44 | 78.30 | 65.08 |
| Qwen2.5-7B-Instruct | QLoRA(RTN) | 3 | 66.84 | 36.45 | 77.53 | **78.83** | 64.91 |
| | LoftQ | | 67.19 | 37.12 | 75.58 | 78.78 | 64.67 |
| | ProjQ | | **74.39** | **42.81** | **77.97** | 78.14 | **68.33** |
| | QLoRA | | **80.70** | **48.16** | 78.24 | **79.26** | **71.59** |
| | LoftQ | 4 | 67.02 | 34.45 | **78.67** | 78.09 | 64.56 |
| | ProjQ | | 69.65 | 40.80 | 78.51 | 78.14 | 66.78 |

*Table 8.* Error compensation results of ProjQ with different values of designed rank ($r_d$) under fixed adaptation rank ($r_a = 64$).

| $r_a$ | $r_d$ | LLaMA-2-7B | | Qwen2.5-7B-Ins | |
|---|---|---|---|---|---|
| | | PPL ($\downarrow$) | Avg. ($\uparrow$) | PPL ($\downarrow$) | Avg. ($\uparrow$) |
| 64 | 16 | 29.07 | 46.85 | 41.04 | 42.91 |
| 64 | 32 | 29.68 | 45.69 | 41.21 | 44.97 |
| **64** | **64** | **22.42** | **47.82** | **38.66** | 44.44 |
| 64 | 96 | 30.42 | 45.70 | 49.26 | 42.03 |
| 64 | 128 | 30.13 | 44.67 | 51.90 | 43.22 |

## C. Experiment Hyperparamter Setup

For LoRA fine-tuning, we show the epoch and learning rate for different settings in Tables 9 to 11.

*Table 9.* Hyperparameters of Language Modeling (WikiText-2) for Llama and Qwen models at different bit widths.

| Parameters | LLAMA-2-7b | | | Qwen2.5-7b-Instruct | | | LLAMA-2-13b | | Qwen2.5-14b-Instruct | |
|---|---|---|---|---|---|---|---|---|---|---|
| | 2bit | 3bit | 4bit | 2bit | 3bit | 4bit | 2bit | 3bit | 2bit | 3bit |
| epoch | 1 | 1 | 1 | 1 | 1 | 1 | 1 | 1 | 1 | 1 |
| lr | 1e-4 | 1e-4 | 1e-4 | 3e-5 | 3e-5 | 3e-5 | 2e-4 | 1e-4 | 3e-4 | 1e-4 |

*Table 10.* Hyperparameters of Mathematical Reasoning (GSM8K) for Llama and Qwen models at different bit widths.

| Parameters | LLAMA-2-7b | | | Qwen2.5-7b-Instruct | | | LLAMA-2-13b | | Qwen2.5-14b-Instruct | |
|---|---|---|---|---|---|---|---|---|---|---|
| | 2bit | 3bit | 4bit | 2bit | 3bit | 4bit | 2bit | 3bit | 2bit | 3bit |
| epoch | 3 | 3 | 3 | 3 | 2 | 3 | 2 | 2 | 2 | 2 |
| lr | 3e-4 | 3e-4 | 3e-5 | 3e-5 | 3e-5 | 3e-5 | 5e-5 | 2e-5 | 5e-5 | 3e-5 |

*Table 11.* Hyperparameters of Common Sense Tasks for Llama and Qwen models at different bit widths.

| Parameters | LLAMA-2-7b | | | Qwen2.5-7b-Instruct | | | LLAMA-2-13b | | Qwen2.5-14b-Instruct | |
|---|---|---|---|---|---|---|---|---|---|---|
| | 2bit | 3bit | 4bit | 2bit | 3bit | 4bit | 2bit | 3bit | 2bit | 3bit |
| epoch | 1 | 1 | 1 | 1 | 1 | 1 | 3 | 3 | 1 | 1 |
| lr | 1e-4 | 1e-4 | 5e-5 | 3e-5 | 3e-5 | 3e-5 | 5e-5 | 5e-5 | 3e-5 | 3e-5 |

