# OpenReview forum: "ProjQ: Project-and-Quantize for Adapter-Aware LLM Compression"
_ICML.cc/2026/Conference — ICML 2026 regular_

### Official Review · Reviewer_ve78 · 2026-03-07

**Soundness:** 4
**Presentation:** 3
**Significance:** 3
**Originality:** 4
**Overall Recommendation:** 4
**Confidence:** 4

**Summary:**

This paper proposes ProjQ (Project-and-Quantize), a novel framework designed to address the structural misalignment between Post-Training Quantization (PTQ) and Low-Rank Adaptation (LoRA) for efficient Large Language Model (LLM) deployment. Traditional PTQ generates isotropic, high-dimensional quantization noise that overwhelms subsequent capacity-limited low-rank adapters, leaving them without sufficient parameter capacity to learn downstream tasks.

To solve this problem, ProjQ formulates the quantization process as a subspace projection problem, mathematically forcing the dominant quantization error into a specific low-rank manifold. The authors derive an alternating optimization algorithm to achieve this noise "shaping" and provide theoretical bounds (Theorem 5.2) proving that, compared to standard PTQ, ProjQ preserves strictly greater model plasticity under task shifts. Empirical evaluations on LLaMA-2 (7B) and Qwen2.5 (7B) demonstrate that ProjQ significantly outperforms existing baseline methods (GPTQ, QLoRA, LoftQ) in both zero-shot error compensation and downstream task fine-tuning. Notably, it enables a 3-bit model to match the performance of standard 4-bit baselines on language modeling tasks.

**Compliance With Llm Reviewing Policy:**

Affirmed.

**Final Justification:**

Overall, this paper introduces a highly original and theoretically well-grounded approach (ProjQ) that elegantly aligns post-training quantization noise with the corrective capacity of low-rank adapters.

During the initial review, my main concerns were the lack of comparisons with recent SOTA LoRA-aware quantization baselines (e.g., QA-LoRA, LQ-LoRA) and the missing systematic ablation studies on rank decoupling ($r_d \neq r_a$).

I thank the authors for their thorough rebuttal. The rebuttal successfully addressed my main concerns by providing the requested baseline comparisons, which further validate ProjQ's superiority. Additionally, the new ablation studies clearly explain the dynamics between the designed rank and adaptation rank. These additions have significantly strengthened the empirical evaluation. Consequently, the rebuttal has positively changed my evaluation, and I am raising my final recommendation to an Accept.

**Key Questions For Authors:**

1. **Missing Baselines:** Could the authors supplement performance comparison experiments between ProjQ and the latest "LoRA-aware quantization baselines" (such as QA-LoRA or LQ-LoRA) mentioned in the related work section? (Or explicitly discuss the expected differences between them).
2. **Rank Decoupling (**$r_d$ **vs.** $r_a$**):** How does the model perform in downstream fine-tuning when $r_d < r_a$ or $r_d > r_a$? For a practitioner with a fixed fine-tuning memory budget (i.e., a fixed $r_a$), what is the principled method or theoretical guidance for selecting the optimal $r_d$?
3. **Scalability:** In larger-scale models (e.g., >70B parameters), as the hidden layer dimensions increase, how will the computational overhead (especially the SVD and covariance matrix operations) scale? Is there a risk of SVD becoming a memory or time bottleneck during the initialization phase?
4. **Time Overhead:** Could the authors provide a wall-clock time comparison for the initialization phase (Phase 1 + Phase 2) of ProjQ versus standard GPTQ and LoftQ initializations, to verify the claim that "the overhead is negligible"?

**Limitations:**

The authors briefly mention future extensions (e.g., non-uniform quantizers) in the conclusion section, but the paper lacks a dedicated "Limitations" or "Broader Impacts" section.

**Constructive suggestion:** The authors should explicitly discuss the limitations faced when scaling their method to larger models (>70B parameters), where the computational overhead of SVD and the alternating optimization loop might not be negligible. Additionally, supplementing a brief statement regarding "the potential societal impact brought by facilitating the widespread deployment of highly compressed LLMs on edge devices" would make this paper more complete.

**Strengths And Weaknesses:**

**Strengths:**

- **Originality:** Rather than merely minimizing the absolute magnitude of the quantization error, approaching it from the perspective of "shaping" the quantization noise is highly innovative. Actively aligning the structure of the quantization error with the corrective capacity of the downstream adapter provides a clever solution to the "dual burden" problem in the PTQ+PEFT pipeline.
- **Soundness:** The theoretical foundation of this paper is exceptionally solid. Proposition 4.1 elegantly transforms the optimization into a subspace projection problem, and Theorem 5.2 provides a rigorous upper bound, proving the superiority of ProjQ under task shifts.
- **Significance:** The empirical results in extreme low-bit (2-bit and 3-bit) scenarios are highly compelling. Achieving downstream performance comparable to 4-bit baselines using only 3 bits highlights the extremely high practical value of this method for resource-constrained edge deployment.

**Weaknesses:**

- **Missing Recent Baselines (Soundness/Significance):** Although the paper mentions recent advances such as QA-LoRA, LQ-LoRA, and LowRA in the related work section, the core experiments only compare ProjQ against standard GPTQ, QLoRA, and LoftQ. The lack of comparison with these latest "LoRA-aware quantization methods" limits our assessment of ProjQ's exact position within the current technological landscape.
- **Limited Model Scale (Significance):** The current evaluation is restricted to models with 7B parameters (LLaMA-2 7B and Qwen2.5-7B). Given the rapid expansion of LLM scales, it is necessary to validate this method on larger models (e.g., 14B, 32B, or 70B) to prove its broad applicability and stability across different architectures.
- **Insufficient Ablation on Rank Decoupling (Soundness):** A key feature of this architecture is the decoupling of the "designed rank" ($r_d$) and the "adaptation rank" ($r_a$). However, identical values ($r_d = r_a = 64$) are used across all downstream task experiments (Tables 2-4). The paper lacks a systematic ablation study to explore the dynamics and performance trade-offs when $r_d \neq r_a$.
- **Missing Concrete Time Overhead Analysis (Presentation):** The paper claims that the computational overhead of SVD operations is negligible during the offline initialization phase. However, considering the presence of the alternating optimization process ($T_{max}=5$), providing a concrete comparison of actual wall-clock time against standard baseline methods would significantly strengthen the persuasiveness of this claim.

---

> ### Author Rebuttal · Authors · 2026-03-31
>
> We thank the reviewer for recognizing the originality of our noise "shaping" approach, the solid theoretical foundation, and the significant practical value of our low-bit results.
>
> **1. Extensive Experiments: Missing Baselines and Model Scalability**
>
> * **Scalability to Larger Models:** We have expanded our evaluation to larger models.
> * **Recent LoRA-Aware Baselines:** We supplemented our core experiments with recent SOTA baselines. ProjQ consistently outperforms these methods. While other methods rely on static truncation or iterative MSE, ProjQ aligns the quantization noise with the LoRA adapter's capacity using subspace projection, preserving strictly greater task plasticity.
>
> **Table 1.** Consolidated results on general NLU benchmarks for LLaMA2 and Qwen2.5 models under 2-bit quantization ($r_a=r_d=64$).
>
> | Model | Method | C4 | WikiText | Avg. Acc |
> |---|---|---|---|---|
> | **Qwen3-8B** | GPTQ+SVD-LLM | 52.65 | 76.19 | 39.55 |
> | | LoftQ | 54.71 | 62.38 | 40.98 |
> | | **ProjQ** | **42.02** | **48.75** | **41.96** |
> | **LLaMA2-13B** | GPTQ+SVD-LLM | 14.50 | 13.76 | 52.97 |
> | | AWQ+SVD-LLM | 9.5e+4 | 1.2e+5 | 37.82 |
> | | LoftQ | 14.14 | 12.80 | 53.82 |
> | | **ProjQ** | **12.48** | **11.56** | **55.58** |
> | **Qwen2.5-14B-Ins** | GPTQ+SVD-LLM | 28.94 | 27.51 | 46.28 |
> | | AWQ+SVD-LLM | NAN | NAN | 36.67 |
> | | LoftQ | 31.74 | 28.10 | 46.10 |
> | | **ProjQ** | **22.22** | **21.50** | **47.68** |
> | **Qwen2.5-32B-Ins** | GPTQ+SVD-LLM | 16.96 | 15.30 | 49.77 |
> | | LoftQ | 16.95 | 15.46 | 53.42|
> | | **ProjQ** | **14.33** | **12.45** | **55.05** |
>
>
> **Table 2.** Fine-tuning results across Common Sense Reasoning, WikiText-2, and GSM8K tasks.
>
> | Task | Bit | Method | LLaMA2-7B | Qwen2.5-7B-Ins | LLaMA2-13B | Qwen2.5-14B-Ins |
> |---|---|---|---|---|---|---|
> | **WikiText-2** | 16 | FP16 | 5.18 | 7.16 | 4.66 | 5.97 |
> | | 2 | LoftQ | 9.14 | 13.15 | 7.33 | 12.17 |
> | | | **ProjQ** | **8.17** | **12.85** | **6.44** | **12.01** |
> | | 3 | LQ-LoRA | 6.73 | - | - | - |
> | | | LoftQ | 6.20 | 8.82 | 5.09 | 7.31 |
> | | | **ProjQ** | **5.69** | **8.02** | **4.98** | **7.08** |
> | **GSM8K** | 16 | FP16 | 38.42 | 70.89 | 46.63 | 82.52 |
> | | 2 | QA-LoRA | 21.30 | - | - | 34.69 |
> | | | LoftQ | 22.52 | 46.85 | 30.98 | **51.25** |
> | | | **ProjQ** | **22.90** | **47.16** | **31.25** | 51.11 |
> | | 3 | LQ-LoRA | 7.40 | - | - | - |
> | | | QA-LoRA | - | - | - | 66.49 |
> | | | LoftQ | **37.68** | 70.02 | 45.03 | 78.39 |
> | | | **ProjQ** | 35.18 | **70.81** | **46.07** | **81.50** |
>
> **2. Ablation on Rank Decoupling ($r_d$ vs $r_a$)**
>
> For a fixed fine-tuning memory budget, selecting the designed rank $r_d$ introduces a key trade-off:
>
> * **If $r_d < r_a$:** This leaves too much unstructured noise in the backbone, underutilizing the adapter's error-correction capacity.
> * **If $r_d > r_a$:** This projects more noise into the subspace than the adapter can absorb, leaving uncorrectable residuals and risking overfitting to the calibration data.
>
>  As shown in Table 3, aligning $r_d$ closely with $r_a$ generally yields the best performance.
>
> **Table 3.** Ablation study on the decoupling of designed rank ($r_d$) and adaptation rank ($r_a$).
>
> | $r_a$ | $r_d$ | LLaMA-2-7B PPL (↓) | LLaMA-2-7B Avg Acc (↑) | Qwen-2.5-7B-Instruct PPL (↓) | Qwen-2.5-7B-Instruct Avg Acc (↑) |
> |:---:|:---:|:---:|:---:|:---:|:---:|
> | 64 | 16 | 29.07 | 46.85 | 41.04 | 42.91 |
> | 64 | 32 | 29.68 | 45.69 | 41.21 | **44.97** |
> | **64** | **64** | **22.42** | **47.82** | **38.66** | 44.44 |
> | 64 | 96 | 30.42 | 45.70 | 49.26 | 42.03 |
> | 64 | 128 | 30.13 | 44.67 | 51.90 | 43.22 |
>
> **3. Scalability, Computational Complexity, and Time Overhead**
>
> To address your questions regarding computational overhead, there is  low risk of SVD or covariance operations bottlenecking larger models. Furthermore, because ProjQ projects noise onto a tightly constrained subspace ($r_d \ll d$), we utilize truncated randomized SVD to extract only the top-$r_d$ components instead of computing the full SVD ($O(d^3)$). **This reduces time complexity to $O(d^2 \cdot r_d)$**, yielding an approximate 30x to 100x theoretical reduction in FLOPs compared to full SVD for a standard layer.
>
> During initialization, ProjQ's computational complexity is highly comparable to LoftQ. While LoftQ relies on the SVD of the residual weight matrix ($W - \widehat{W}$), our covariance-aware initialization (Lemma 4.2) achieves similar theoretical efficiency. **Activation-based SVD typically requires dense matrix inversions, but our formulation strictly inverts a diagonal matrix, avoiding standard computational bottlenecks.** Consequently, the empirical wall-clock times for initializing LLaMA2-13B are highly comparable (**13 min for LoftQ vs. 17 min for ProjQ**). This minor difference is brought entirely by the necessary computation of the activation covariance matrices, which is computationally inexpensive and does not pose a significant delay. This one-time offline cost represents less than $10\%$ of the total fine-tuning time.

---

> > ### Author Rebuttal · Reviewer_ve78 · 2026-04-03
> >
> > **Thank you for the comprehensive and highly convincing rebuttal.**
> >
> > The authors have done a phenomenal job addressing all of my primary concerns.
> >
> > 1. **Baselines & Scalability:** I deeply appreciate the extensive effort to include Qwen2.5-32B and LLaMA2-13B models, as well as the addition of SOTA baselines (QA-LoRA, LQ-LoRA, SVD-LLM). The fact that ProjQ continues to outperform these recent strong baselines at the 32B scale significantly bolsters the empirical claims of the paper.
> > 2. **Rank Decoupling:** The ablation study on $r_d$ vs. $r_a$ (Table 3) and the accompanying theoretical explanation perfectly clarify the trade-off mechanics.
> > 3. **Computational Overhead:** The clarification regarding the use of Truncated Randomized SVD ($O(d^2\cdot r_d)$) is crucial and completely resolves my concerns about $O(d^3)$ bottlenecks. The wall-clock time comparison (17m vs 13m) provides the exact concrete
> > evidence I requested.
> >
> > Because the authors have rigorously closed the empirical gaps identified in the initial review, I am raising my Overall Recommendation from Weak Reject (3) to Weak Accept (4), and increasing my Soundness score from Good (3) to Excellent (4).
> >
> > **Minor suggestion for the Camera-Ready version:** Please ensure that the details regarding the Truncated Randomized SVD (and its complexity analysis), the wall-clock time comparisons, and the $r_d$ vs. $r_a$ ablation are prominently incorporated into the main text or a dedicated appendix section. These details are highly valuable for practitioners looking to reproduce or deploy this
> > framework.
> >
> > Great work!

---

> > > ### Author Response · Authors · 2026-04-06
> > >
> > > We sincerely thank the reviewer for the thorough evaluation and for the positive shift in both the overall score and the soundness rating. We are particularly pleased that our additional experiments on Qwen2.5-32B and the complexity analysis of the Truncated Randomized SVD successfully addressed your concerns regarding scalability and overhead. We fully commit to incorporating these new empirical results, the wall-clock comparisons, and the rank-decoupling ablation into the final version as suggested.

---

### Official Review · Reviewer_jW9D · 2026-03-10

**Soundness:** 4
**Presentation:** 2
**Significance:** 3
**Originality:** 3
**Overall Recommendation:** 4
**Confidence:** 4

**Summary:**

This paper proposes a new perspective that PTQ and lora finetuning should be co-desinged. the motivation is that traditional quantization-base lora finetuning may waste capacity on fixing quantizaiton noise. to address the issue, this paper derives an alternating algorithm that shapes the quantization noise into a low-rank structure, offloading dominant error components to the subsequent adapter while minimizing the residual error in the orthogonal ”uncorrectable” subspace. results on llama2-7b and qwen2.5-7b show notable improvement.

**Compliance With Llm Reviewing Policy:**

Affirmed.

**Final Justification:**

The rebuttal addressed most of my concerns and strengthened the paper through additional experiments and clarifications. Some presentation and completeness issues remain. I therefore maintain my WA recommendation.

**Key Questions For Authors:**

see weakness

**Limitations:**

authors did not discuss the limitations and potential negative societal impact.

**Strengths And Weaknesses:**

### Strengths
- the motivation that projects quant error to low rank subspace is novel
- the formulations are well designed and make sense


### Weaknesses
- this paper is not easy to follow. authors should consider adding at least one figure to illustrate the main idea and pipeline.
- more results on advanced and larger models are preferred. however, this paper only evaluates on llama2-7b and qwen2.5-7b
- fp16 baselines are not provided, so that it is confused for me to determine whether the 2bit quantization results are able to use.
- there are several existing studies focusing on "Quantization Error Compensation", such as SVDQuant, CALDERA, FBQuant, which should be considered as baseline methods

---

> ### Author Rebuttal · Authors · 2026-03-31
>
> We sincerely thank the reviewer for recognizing the novelty of our approach and the soundness of our formulation. We greatly appreciate your constructive feedback. Our answers are as follows.
>
> **1. Main Idea and Pipeline Illustration**
>
> We agree that the paper would benefit from a clearer illustration of the overall pipeline. To address this, we will include a comprehensive system figure in the revised version. Below is a structured overview of the ProjQ framework, illustrating how PTQ and LoRA fine-tuning are mathematically co-designed into two streamlined phases:
>
>
>     [ Input: Pre-trained LLM Weights (W) ]
>
>     Phase 1: Alternating Subspace Projection (Offline PTQ)
>     --------------------------------------------------------------------------------
>     Goal: Quantize W while forcing the quantization noise into a low-rank subspace P.
>       * Step A: Fix subspace P, update quantized weights W_hat.
>       * Step B: Fix W_hat, update subspace P to capture the dominant error.
>       * Step C: Repeat Step A and Step B until convergence.
>       --> Output: Quantized weights (W_hat) and Structured Low-Rank Noise.
>
>     Phase 2: Initialization + LoRA-Based PEFT for Training
>     --------------------------------------------------------------------------------
>     Goal: Maximally cancel structured noise at step zero, then adapt to the new task.
>       * Step A (Initialization): Set LoRA matrices (A, B) such that B*A mathematically
>         mirrors and maximally cancels the low-rank error captured in Phase 1's subspace P.
>       * Step B (Fine-Tuning): Train the pre-initialized adapter.
>
>
> **2. Extensive Experiments: FP16, Larger Models, and Recent Baselines**
>
> To comprehensively validate the efficiency and usability of ProjQ, we have significantly scaled our experiments in multiple directions:
>
> * **FP16 Upper Bound:** We have included the full-precision (FP16) results in finetuning results.
> * **Scaling to Larger Models:** We have extended our evaluations to include larger models. ProjQ's effectiveness proves to be highly stable and scalable on these massive architectures, proving our method is not limited to 7B parameter models.
> * **Quantization Error Compensation Baselines:** We have included more recent state-of-the-art baselines that focus on error compensation. ProjQ consistently outperforms these methods by ensuring the residual noise is structurally aligned with the downstream adapter, preserving greater "task plasticity."
>
> **Table 1.** Consolidated results on general NLU benchmarks for LLaMA2 and Qwen2.5 models under 2-bit quantization ($r_a=r_d=64$). Perplexity (↓) and Accuracy (↑, %) are reported.
>
> | Model | Method | C4 | WikiText | Avg. Acc |
> |---|---|---|---|---|
> | **LLaMA2-7B** | GPTQ+SVD-LLM | 26.26 | 28.81 | 45.33 |
> | | AWQ+SVD-LLM | 1.7e+5 | 2.2e+5 | 36.61 |
> | | CALDERA | 21.59 | 23.82 | 46.85 |
> | | **ProjQ** | **21.50** | **22.42** | **47.82** |
> | **Qwen2.5-7B-Ins** | GPTQ+SVD-LLM | 62.27 | 92.21 | 40.96 |
> | | AWQ+SVD-LLM | NAN | NAN | 37.51 |
> | | CALDERA | 50.57 | 70.52 | 42.50 |
> | | **ProjQ** | **33.50** | **38.66** | **44.44** |
> | **Qwen3-8B** | GPTQ+SVD-LLM | 52.65 | 76.19 | 39.55 |
> | | LoftQ | 54.71 | 62.38 | 40.98 |
> | | **ProjQ** | **42.02** | **48.75** | **41.96** |
> | **LLaMA2-13B** | GPTQ+SVD-LLM | 14.50 | 13.76 | 52.97 |
> | | AWQ+SVD-LLM | 9.5e+4 | 1.2e+5 | 37.82 |
> | | LoftQ | 14.14 | 12.80 | 53.82 |
> | | **ProjQ** | **12.48** | **11.56** | **55.58** |
> | **Qwen2.5-14B-Ins** | GPTQ+SVD-LLM | 28.94 | 27.51 | 46.28 |
> | | AWQ+SVD-LLM | NAN | NAN | 36.67 |
> | | LoftQ | 31.74 | 28.10 | 46.10 |
> | | **ProjQ** | **22.22** | **21.50** | **47.68** |
> | **Qwen2.5-32B-Ins** | GPTQ+SVD-LLM | 16.96 | 15.30 | 49.77 |
> | | LoftQ | 16.95 | 15.46 |53.42 |
> | | **ProjQ** | **14.33** | **12.45** | **55.05** |
>
>
> **Table 2.** Fine-tuning results across Common Sense Reasoning, WikiText-2, and GSM8K tasks.
>
> | Task | Bit | Method | LLaMA2-7B | Qwen2.5-7B-Ins | LLaMA2-13B | Qwen2.5-14B-Ins |
> |---|---|---|---|---|---|---|
> | **Common Sense** | 16 | FP16 | 68.07 | 69.84 | 70.61 | 73.44 |
> | | 2 | LoftQ | 54.78 | 58.86 | 60.25 | 59.08 |
> | | | **ProjQ** | **57.59** | **59.88** | **61.84** | **59.85** |
> | | 3 | LoftQ | 66.38 | 64.67 | 69.10 | 69.89 |
> | | | **ProjQ** | **67.13** | **68.33** | **69.91** | **71.61** |
> | **WikiText-2** | 16 | FP16 | 5.18 | 7.16 | 4.66 | 5.97 |
> | | 2 | LoftQ | 9.14 | 13.15 | 7.33 | 12.17 |
> | | | **ProjQ** | **8.17** | **12.85** | **6.44** | **12.01** |
> | | 3 | LQ-LoRA | 6.73 | - | - | - |
> | | | LoftQ | 6.20 | 8.82 | 5.09 | 7.31 |
> | | | **ProjQ** | **5.69** | **8.02** | **4.98** | **7.08** |
> | **GSM8K** | 16 | FP16 | 38.42 | 70.89 | 46.63 | 82.52 |
> | | 2 | QA-LoRA | 21.30 | - | - | 34.69 |
> | | | LoftQ | 22.52 | 46.85 | 30.98 | **51.25** |
> | | | **ProjQ** | **22.90** | **47.16** | **31.25** | 51.11 |
> | | 3 | LQ-LoRA | 7.40 | - | - | - |
> | | | QA-LoRA | - | - | - | 66.49 |
> | | | LoftQ | **37.68** | 70.02 | 45.03 | 78.39 |
> | | | **ProjQ** | 35.18 | **70.81** | **46.07** | **81.50** |

---

> > ### Author Rebuttal · Reviewer_jW9D · 2026-04-03
> >
> > Thank you for the detailed rebuttal. The additional experiments resolve most of my concerns and strengthen the paper. However, I believe many of these important results should have been included in the main paper rather than only in the rebuttal. Overall, the concerns are largely addressed, but the paper still needs further polishing in presentation and completeness. I therefore maintain WA.

---

> > > ### Author Response · Authors · 2026-04-06
> > >
> > > We thank the reviewer for the positive assessment. We are pleased that our rebuttal experiments and clarifications successfully resolved the majority of your concerns and strengthened the overall contribution of the paper. We completely agree that the additional results, specifically the Qwen2.5-32B benchmarks, the SVD-LLM comparison, and the rank-decoupling ablation, are essential for a complete presentation of ProjQ. We fully commit to integrating these findings into the final version.
> > >
> > > Furthermore, we will take this opportunity to thoroughly polish the manuscript's presentation, ensuring that the theoretical insights regarding the dual burden are seamlessly balanced with the new empirical evidence. Thank you again for the valuable feedback that has helped us improve the completeness of this work.

---

### Official Review · Reviewer_TcDn · 2026-03-12

**Soundness:** 3
**Presentation:** 3
**Significance:** 3
**Originality:** 3
**Overall Recommendation:** 4
**Confidence:** 4

**Summary:**

This paper proposes ProjQ, a projection-based framework that aligns post-training quantization with subsequent low-rank adaptation for large language models. The key idea is to shape quantization noise so that it lies in a correctable low-rank subspace, thereby improving the effectiveness of downstream adapter fine-tuning. The authors formulate the problem using an activation-aware objective and develop an alternating optimization algorithm that jointly updates quantized weights and error subspaces. They further provide a covariance-aware initialization scheme for low-rank adapters to reduce initial performance degradation after quantization. Empirical results on several LLM backbones show consistent, though moderate, improvements over existing quantization and adaptation baselines.

**Compliance With Llm Reviewing Policy:**

Affirmed.

**Final Justification:**

Thank you for the additional rebuttal. The authors have addressed most of my concerns, so I have raised my score to WA.

**Key Questions For Authors:**

Please see the weaknesses section.

**Limitations:**

No; the paper does not appear to meaningfully discuss its limitations or any potential negative societal impact, so the authors should add a brief section covering expected failure modes, sensitivity across architectures/scales.

**Strengths And Weaknesses:**

[Strengths]

(S1) The paper clearly articulates a limitation of the standard PTQ-then-LoRA pipeline, namely that unstructured quantization noise can consume the limited capacity of low-rank adapters. Framing the problem in terms of preserving correctability during quantization is conceptually sound and provides a strong motivation for the proposed method.

(S2) The paper provides a useful theoretical treatment of the projection-based formulation and the alternating optimization procedure. This analysis helps justify the design of the method and makes the overall framework more principled than a purely heuristic combination of quantization and adapter initialization.

[Weaknesses]

(W1) The evaluation appears to focus on relatively limited model families, and it remains unclear whether the same conclusions hold for more recent architectures and a broader range of scales, such as Qwen3 4B, 8B, or 32B. This is particularly relevant because the reported gains seem to vary noticeably across architectures, suggesting that the effectiveness of the method may be architecture-dependent.

(W2) While the method is technically well motivated, the experimental gains do not always appear sufficiently strong to clearly establish a compelling practical advantage over existing baselines. As a result, the empirical section feels more incremental than decisive in its current form.

(W3) Since the method consists of two key components—Phase 1 for shaping the quantization error and Phase 2 for adapter initialization—it would be important to isolate their individual contributions. For example, it would be useful to evaluate the performance obtained after Phase 1 when using a standard LoRA initialization instead of the proposed initialization scheme. Such an ablation would clarify whether both phases are necessary and how much each component contributes to the final gains.

---

> ### Author Rebuttal · Authors · 2026-03-31
>
> We thank the reviewer for recognizing our formulation as conceptually sound and principled. Please find our answers.
>
> **1. Extensive Evaluation on Larger Models and Baselines**
>
> We validated this with new simulations on Qwen3-8B, Llama2-13B, Qwen2.5-14B, and Qwen2.5-32B. ProjQ yields significant gains over GPTQ and remains superior against other advanced methods, as shown in Tables 1 and 2.
>
> **Table 1.** Consolidated results on general NLU benchmarks for LLaMA2 and Qwen2.5 models under 2-bit quantization ($r_a=r_d=64$).
>
> | Model | Method | C4 | WikiText | Avg. Acc |
> |---|---|---|---|---|
> | **LLaMA2-7B** | GPTQ+SVD-LLM | 26.26 | 28.81 | 45.33 |
> | | AWQ+SVD-LLM | 1.7e+5 | 2.2e+5 | 36.61 |
> | | CALDERA | 21.59 | 23.82 | 46.85 |
> | | **ProjQ** | **21.50** | **22.42** | **47.82** |
> | **Qwen2.5-7B-Ins** | GPTQ+SVD-LLM | 62.27 | 92.21 | 40.96 |
> | | AWQ+SVD-LLM | NAN | NAN | 37.51 |
> | | CALDERA | 50.57 | 70.52 | 42.50 |
> | | **ProjQ** | **33.50** | **38.66** | **44.44** |
> | **Qwen3-8B** | GPTQ+SVD-LLM | 52.65 | 76.19 | 39.55 |
> | | LoftQ | 54.71 | 62.38 | 40.98 |
> | | **ProjQ** | **42.02** | **48.75** | **41.96** |
> | **Qwen2.5-14B-Ins** | GPTQ+SVD-LLM | 28.94 | 27.51 | 46.28 |
> | | AWQ+SVD-LLM | NAN | NAN | 36.67 |
> | | LoftQ | 31.74 | 28.10 | 46.10 |
> | | **ProjQ** | **22.22** | **21.50** | **47.68** |
> | **Qwen2.5-32B-Ins** | GPTQ+SVD-LLM | 16.96 | 15.30 | 49.77 |
> | | LoftQ | 16.95 | 15.46 |53.42 |
> | | **ProjQ** | **14.33** | **12.45** | **55.05** |
>
>
> **Table 2.** Fine-tuning results across Common Sense Reasoning, WikiText-2, and GSM8K tasks.
>
> | Task | Bit | Method | LLaMA2-7B | Qwen2.5-7B-Ins | LLaMA2-13B | Qwen2.5-14B-Ins |
> |---|---|---|---|---|---|---|
> | **WikiText-2** | 16 | FP16 | 5.18 | 7.16 | 4.66 | 5.97 |
> | | 2 | LoftQ | 9.14 | 13.15 | 7.33 | 12.17 |
> | | | **ProjQ** | **8.17** | **12.85** | **6.44** | **12.01** |
> | | 3 | LQ-LoRA | 6.73 | - | - | - |
> | | | LoftQ | 6.20 | 8.82 | 5.09 | 7.31 |
> | | | **ProjQ** | **5.69** | **8.02** | **4.98** | **7.08** |
> | **GSM8K** | 16 | FP16 | 38.42 | 70.89 | 46.63 | 82.52 |
> | | 2 | QA-LoRA | 21.30 | - | - | 34.69 |
> | | | LoftQ | 22.52 | 46.85 | 30.98 | **51.25** |
> | | | **ProjQ** | **22.90** | **47.16** | **31.25** | 51.11 |
> | | 3 | LQ-LoRA | 7.40 | - | - | - |
> | | | QA-LoRA | - | - | - | 66.49 |
> | | | LoftQ | **37.68** | 70.02 | 45.03 | 78.39 |
> | | | **ProjQ** | 35.18 | **70.81** | **46.07** | **81.50** |
>
> **2. Analyzing Performance Gains and Future Integrations**
>
> ProjQ's performance advantages vary across architectures and tasks. This might be explained as follows:
> * **Spectral entropy:** ProjQ's efficacy depends on the spectral decay of the weight-activation product $WX$. Lower entropy often means noise is highly compressible into our low-rank subspace. For instance, **Qwen2.5-7B-INS has a lower entropy (6.89)  vs Llama2-7B's entropy (7.39)**, allowing ProjQ to achieve a relatively greater perplexity reduction on Qwen2.5-7B-INS.*
> * **Model Shift:** Tasks like WikiText-2 require a smaller model shift than GSM8K. Since PTQ lacks downstream task info, we can only align the noise *structure*. A smaller shift allows the adapter to absorb this structured noise without exhausting its capacity, yielding better relative performance.
>
> While ProjQ consistently outperforms existing baselines, we acknowledge that the performance gap is not always massive across every single metric. However, the true strength of ProjQ lies in its modularity and orthogonality to existing literature.
>
> Crucially, while existing advanced quantization approaches successfully minimize the *magnitude* of quantization error, they do not attempt to *shape* this residual noise into a correctable low-rank subspace. Because these two objectives are completely complementary, more advanced methods can be directly implemented into our current framework to unlock compounding, synergistic gains. **Without altering the fundamental architecture of ProjQ, we can significantly boost its efficiency simply by upgrading the implementation of its intermediate blocks.** One interesting enhancement can be **swapping the core quantization solver:** Replacing GPTQ in Algorithm 1 with methods that incorporate linear transformations into the weighting matrix (e.g., QuIP) or introducing non-uniform/mixed-precision quantizers (e.g., AWQ, LowRA).
>
> **3. Ablation on Phase 1 vs. Phase 2**
>
> To isolate our framework's components, we compared Phase 1 using standard LoRA initialization versus our Covariance-Aware initialization. These results demonstrate the efficiency of our initialization.
>
> **Table 3.** Ablation isolating Phase 2 contribution.
>
> | Model | Initialization Scheme | WikiText-2 PPL (↓) | Commonsense Avg Acc (↑) |
> |---|---|---|---|
> | **LLaMA-2-7B** | ProjQ + Standard LoRA Init (Random) | 9.17 | 56.69|
> | | **ProjQ + Proposed Init (Ours)** | **8.17** | **57.59** |
> | **Qwen-2.5-7B-Instruct** | ProjQ + Standard LoRA Init (Random) | 16.40 | 53.66 |
> | | **ProjQ + Proposed Init (Ours)** | **12.85** | **59.88** |

---

> > ### Author Rebuttal · Reviewer_TcDn · 2026-04-03
> >
> > I thank the authors for the detailed rebuttal and for their clear effort to address my concerns. I appreciate that the authors expanded the evaluation to additional and larger models, which partially alleviates my concern about limited model coverage. I also appreciate the added ablation comparing the standard LoRA initialization with the proposed covariance-aware initialization, which helps clarify the contribution of Phase 2 and substantially addresses that point. However, I remain only partially convinced on the generalization issue, since the rebuttal still does not fully clarify whether the method behaves consistently across more recent architectures and broader within-family scales, especially given the variation in gains across model families (e.g., Qwen3-4B and 32B). More importantly, while the discussion on spectral entropy, model shift, and future integration with stronger quantization solvers is interesting, it does not fully resolve my main empirical concern that the practical gains, in their current form, still appear somewhat incremental rather than clearly decisive. Therefore, while I acknowledge that some of my concerns have been addressed, I will maintain my current score.

---

> > > ### Author Response · Authors · 2026-04-07
> > >
> > > We thank the reviewer for acknowledging that our rebuttal addressed the concerns regarding model coverage and the specific contributions of our Phase 2 covariance-aware initialization.
> > >
> > > Regarding the performance across more recent architectures and within-family scales, we have tested the zero-shot performance on the Qwen3 family (4B, 8B, and 32B). As shown below, ProjQ consistently outperforms the baselines. Our experiments demonstrate that ProjQ brings reliable gains compared to existing approaches across different architectures (Llama 2, Qwen 2.5, Qwen 3) and various tasks (zero-shot performance, Commonsense tasks, Wikitext2, and GSM8K).
> > >
> > > | Model | Method | C4 | WikiText | Avg. Acc |
> > > | :--- | :--- | :--- | :--- | :--- |
> > > | **Qwen3-4B** | GPTQ+SVD-LLM | 88.43 | 116.56 | 38.38 |
> > > | | LoftQ | 105.34 | 140.70 | 38.81 |
> > > | | **ProjQ** | **83.64** | **101.22** | **39.28** |
> > > | **Qwen3-8B** | GPTQ+SVD-LLM | 52.65 | 76.19 | 39.55 |
> > > | | LoftQ | 54.71 | 62.38 | 40.98 |
> > > | | **ProjQ** | **42.02** | **48.75** | **41.96** |
> > > | **Qwen3-32B** | GPTQ+SVD-LLM | 26.24 | 33.69 | 44.09 |
> > > | | LoftQ | 25.71 | 33.01 | 44.88 |
> > > | | **ProjQ** | **20.74** | **21.18** | **47.23** |
> > >
> > > Regarding the magnitude of the empirical gains, we respectfully argue the following:
> > >
> > > *  It is important to note that the merit of our work is not limited to empirical simulations; it also lies in the discovery of the dual burden (identifying why adapters struggle on noisy backbones), the provision of a theoretically-grounded and improvable scheme to mitigate it, and the derivation of analytical dominance and explainability results. Nonetheless, the simulations have been enriched to further support our findings.
> > >
> > >
> > > *  The regime of ultra-low bit quantization is known to be challenging. In this context, using low-complexity approaches, a consistent improvement of even a few percent （e.g., 20% improvement for Qwen3 models as shown in the table above） represents a significant bridge toward closing the gap with full-precision performance.
> > >
> > > *  Because ProjQ is an offline initialization method with negligible computational overhead, these gains are achieved at essentially very low cost compared to training-heavy alternatives.
> > >
> > > * On the generalization across scales, the observed variation actually supports our theoretical framework. As models scale, their weight matrices exhibit more pronounced spectral decay and lower intrinsic dimensionality relative to their size. This allows ProjQ to more effectively hide quantization noise in the redundant subspace of larger models. Thus, the variation is not a failure of consistency, but a predictable outcome of model physics.
> > >
> > > We therefore believe the identified dual burden provides a novel and decisive insight into the interaction between discrete backbones and continuous adapters, offering a scalable foundation for the next generation of PEFT-aware quantization.

---

### Official Review · Reviewer_pL89 · 2026-03-12

**Soundness:** 2
**Presentation:** 2
**Significance:** 2
**Originality:** 2
**Overall Recommendation:** 4
**Confidence:** 3

**Summary:**

This paper introduce novel ProjQ, a framework that mathematically structures this quantization noise so the adapter can easily correct it, resulting in better downstream performance and higher efficiency.

**Compliance With Llm Reviewing Policy:**

Affirmed.

**Final Justification:**

Both of the major concerns are resolved by authors, I have increased the score, considering rebuttal.

**Key Questions For Authors:**

Same as in Weakness section.

**Limitations:**

yes

**Strengths And Weaknesses:**

Strength:

This paper is well written and organized beautifully, with easy language. The proposed work is quite novel and provide a theoretical background on the proposed idea.

This work confines quantization noise to a low-rank subspace, allowing the subsequent adapter to correct it.


Weakness and Questions:

1. In the current world, 7B parameters are considered quite small. This article should scale their experiments up to larger models (e.g., 13B, 30B, or 70B parameters), this will help us to proof that the proposed method is  stable or effective on truly massive architectures.

2. To strengthen the empirical claims, the experimental section must evaluate ProjQ against a broader set of recent, state-of-the-art quantization-aware fine-tuning baselines (i.e. SVD-LLM, AWQ, others) , rather than relying solely on foundational methods like GPTQ, QLoRA, and LoftQ."

---

> ### Author Rebuttal · Authors · 2026-03-31
>
> We sincerely thank the reviewer for the constructive feedback and encouraging comments concerning our novel core approach, which is to geometrically shape quantization noise into a correctable low-rank subspace; and this with low complexity.  We address your main points below to demonstrate ProjQ's robustness.
>
> **1. Evaluation on larger models**
>
> While a primary motivation of our work (Sec. 1) aligns with the growing community focus on deploying (small) LLMs on-device, scalability is indeed crucial.
>
> First, several of our theoretical/explainability/constructive results hold for all model sizes: Prop. 4.1 establishes a geometric equivalence that is independent of specific $m$ or $n$ values, and Theo. 5.2 guarantees that ProjQ will dominate classical PTQ methods regardless of the number of parameters. The mechanisms of subspace projection and alternating optimization operate layer-wise, making them naturally scalable.
>
> Regarding performance gains, larger models typically exhibit lower intrinsic dimensionality relative to their total parameter count (see e.g., Aghajanyan et al ACL2021). This implies that the low-rank assumption for task shifts is arguably more valid.
> %Therefore, the competition for adapter capacity, in which adapters must simultaneously fix quantization noise and learn new tasks, is a universal problem that is even more fierce for larger models.
>
>  To validate this, we have run new simulations on Qwen3-8B, LLaMA2-13B, Qwen2.5-14B, Qwen2.5-32B. As shown in the tables below, **ProjQ consistently outperforms existing methods for different architectures and sizes, since the core idea of ProjQ by shaping the quantization noise into a subspace is independent of the model size and architecture.**
>
> **2. Extended comparisons with recent baselines**
>
> We have expanded our evaluation to include more baselines such as **SVD-LLM and AWQ**. Several remarks are in order: Using salience-aware quantization schemes to replace GPTQ in Algo.~1 would be a relevant extension of ProjQ; ProjQ can be seen as a generalized evolution of the SVD-LLM concept. **While SVD-LLM focuses on static low-rank truncation, ProjQ extends this by introducing a dynamic, activation-aware projection.**
>
> As demonstrated in the tables below, structuring the quantization error to fit the adapter's capacity effectively preserves the model's downstream learning ability. **Consequently, ProjQ yields clear performance advantages over these recent baselines in low bit regimes.**
>
>
> **Table 1.** Consolidated results on general NLU benchmarks for LLaMA2 and Qwen2.5 models under 2-bit quantization ($r_a=r_d=64$).
>
> | Model | Method | C4 | WikiText | Avg. Acc |
> |---|---|---|---|---|
> | **LLaMA2-7B** | GPTQ+SVD-LLM | 26.26 | 28.81 | 45.33 |
> | | AWQ+SVD-LLM | 1.7e+5 | 2.2e+5 | 36.61 |
> | | CALDERA | 21.59 | 23.82 | 46.85 |
> | | **ProjQ** | **21.50** | **22.42** | **47.82** |
> | **Qwen2.5-7B-Ins** | GPTQ+SVD-LLM | 62.27 | 92.21 | 40.96 |
> | | AWQ+SVD-LLM | NAN | NAN | 37.51 |
> | | CALDERA | 50.57 | 70.52 | 42.50 |
> | | **ProjQ** | **33.50** | **38.66** | **44.44** |
> | **Qwen3-8B** | GPTQ+SVD-LLM | 52.65 | 76.19 | 39.55 |
> | | LoftQ | 54.71 | 62.38 | 40.98 |
> | | **ProjQ** | **42.02** | **48.75** | **41.96** |
> | **LLaMA2-13B** | GPTQ+SVD-LLM | 14.50 | 13.76 | 52.97 |
> | | AWQ+SVD-LLM | 9.5e+4 | 1.2e+5 | 37.82 |
> | | LoftQ | 14.14 | 12.80 | 53.82 |
> | | **ProjQ** | **12.48** | **11.56** | **55.58** |
> | **Qwen2.5-14B-Ins** | GPTQ+SVD-LLM | 28.94 | 27.51 | 46.28 |
> | | AWQ+SVD-LLM | NAN | NAN | 36.67 |
> | | LoftQ | 31.74 | 28.10 | 46.10 |
> | | **ProjQ** | **22.22** | **21.50** | **47.68** |
> | **Qwen2.5-32B-Ins** | GPTQ+SVD-LLM | 16.96 | 15.30 | 49.77 |
> | | LoftQ | 16.95 | 15.46 | 53.42 |
> | | **ProjQ** | **14.33** | **12.45** | **55.05** |
>
>
> **Table 2.** Fine-tuning results across Common Sense Reasoning, WikiText-2, and GSM8K tasks.
>
> | Task | Bit | Method | LLaMA2-7B | Qwen2.5-7B-Ins | LLaMA2-13B | Qwen2.5-14B-Ins |
> |---|---|---|---|---|---|---|
> | **Common Sense** | 16 | FP16 | 68.07 | 69.84 | 70.61 | 73.44 |
> | | 2 | LoftQ | 54.78 | 58.86 | 60.25 | 59.08 |
> | | | **ProjQ** | **57.59** | **59.88** | **61.84** | **59.85** |
> | | 3 | LoftQ | 66.38 | 64.67 | 69.10 | 69.89 |
> | | | **ProjQ** | **67.13** | **68.33** | **69.91** | **71.61** |
> | **WikiText-2** | 16 | FP16 | 5.18 | 7.16 | 4.66 | 5.97 |
> | | 2 | LoftQ | 9.14 | 13.15 | 7.33 | 12.17 |
> | | | **ProjQ** | **8.17** | **12.85** | **6.44** | **12.01** |
> | | 3 | LQ-LoRA | 6.73 | - | - | - |
> | | | LoftQ | 6.20 | 8.82 | 5.09 | 7.31 |
> | | | **ProjQ** | **5.69** | **8.02** | **4.98** | **7.08** |
> | **GSM8K** | 16 | FP16 | 38.42 | 70.89 | 46.63 | 82.52 |
> | | 2 | QA-LoRA | 21.30 | - | - | 34.69 |
> | | | LoftQ | 22.52 | 46.85 | 30.98 | **51.25** |
> | | | **ProjQ** | **22.90** | **47.16** | **31.25** | 51.11 |
> | | 3 | LQ-LoRA | 7.40 | - | - | - |
> | | | QA-LoRA | - | - | - | 66.49 |
> | | | LoftQ | **37.68** | 70.02 | 45.03 | 78.39 |
> | | | **ProjQ** | 35.18 | **70.81** | **46.07** | **81.50** |

---

> > ### Author Rebuttal · Reviewer_pL89 · 2026-04-04
> >
> > Thanks for the author's response. My most issues have been dealt, and I will increase my score.

---

> > > ### Author Response · Authors · 2026-04-06
> > >
> > > We sincerely appreciate your support of our rebuttal and your decision to raise the score.

---

### Decision · Program_Chairs · 2026-04-30

**Decision:**

Accept (regular)

**Comment:**

The paper presents a novel and technically well-motivated framework for aligning post-training quantization with low-rank adaptation. Reviewers generally agreed on the originality of the core idea and found the rebuttal effective in addressing the main empirical concerns, including larger-scale evaluation, stronger baselines, additional ablations, and overhead clarification. Based on the conceptual contribution, sound methodology, and strengthened evidence after rebuttal, I recommend acceptance.

For the final manuscript, the authors should focus on a small number of high-impact revisions: incorporate the key rebuttal results directly into the paper, especially the larger-model evaluations and stronger baseline comparisons; add a concise ablation clarifying the contributions of the two phases; and improve clarity with a simple overview figure and a brief limitations discussion. This would make the paper more complete and easier for readers to assess and reproduce.